# Meta Guidance: Incorporating Inductive Biases into Deep Time Series Imputers

**Jiacheng You[1]    Xinyang Chen[1✉]    Yu Sun[2]    Weili Guan[3]    Liqiang Nie[1]**

[1]School of Computer Science and Technology, Harbin Institute of Technology (Shenzhen)
[2]College of Computer Science, DISSec, Nankai University
[3]School of Information Science and Technology, Harbin Institute of Technology (Shenzhen)
{jiachengyou2hit, chenxinyang95, nieliqiang}@gmail.com
sunyu@nankai.edu.cn, guanweili@hit.edu.cn

## Abstract

Missing values, frequently encountered in time series data, can significantly impair the effectiveness of analytical methods. While deep imputation models have emerged as the predominant approach due to their superior performance, explicitly incorporating inductive biases aligned with time-series characteristics offers substantial improvement potential. Taking advantage of non-stationarity and periodicity in time series, two domain-specific inductive biases are designed: (1) Non-Stationary Guidance, which operationalizes the proximity principle to address highly non-stationary series by emphasizing temporal neighbors, and (2) Periodic Guidance, which exploits periodicity patterns through learnable weight allocation across historical periods. Building upon these complementary mechanisms, the overall module, named Meta Guidance, dynamically fuses both guidances through data-adaptive weights learned from the specific input sample. Experiments on nine benchmark datasets demonstrate that integrating Meta Guidance into existing deep imputation architectures achieves an average 27.39% reduction in imputation error compared to state-of-the-art baselines.

## 1    Introduction

Multivariate time series data are ubiquitous in domains ranging from finance to healthcare. However, real-world deployments commonly exhibit missing values [32, 43] caused by sensor malfunctions, communication disruptions, or incomplete data acquisition. Such missingness undermines data integrity and impedes downstream applications [11], establishing accurate imputation as a critical prerequisite for robust modeling and analysis.

Traditional imputation methods [24, 13] have demonstrated competence in time series completion. The deep learning revolution has further advanced imputation techniques [5, 4], capitalizing on their ability to model nonlinear dependencies. Recent innovations in diffusion models [34] and transformer architectures [41, 8] have emerged as state-of-the-art imputers. However, current approaches always adopt end-to-end learning to implicitly infer temporal patterns, largely overlooking the strategic integration of domain-specific inductive biases known to enhance imputation. This motivates our core research problem: *Can deep time series imputers achieve superior performance by incorporating domain-specific inductive biases rather than relying exclusively on data-driven pattern discovery?*

In this paper, we introduce an intuitive and interpretable approach for embedding inductive biases into deep time series imputation models. Real-world time series often exhibit non-stationary trends [14] and periodicity [9]. To capture these properties, we design two interpretable guidance matrices: **Non-Stationary Guidance (NSG)** and **Periodic Guidance (PG)**. In the NSG matrices, the "proximity principle" is employed, wherein heightened importance is assigned to values in close proximity to the

39th Conference on Neural Information Processing Systems (NeurIPS 2025).

missing value. In the PG matrices, increased importance is assigned to values located at one or more period lengths away from the missing value. Recognizing that different segments of time series data vary in their dominant characteristics, we develop **Meta Guidance (MG)** — a learnable mechanism that adaptively fuses NSG and PG based on specific input. When injected into advanced deep architectures (TCN, diffusion, and Transformer), MG consistently improves performance, yielding a 27.39% average error reduction across nine datasets.

Our main contributions are summarized as follows:

- By exploiting the *non-stationarity* and *periodicity* properties of time series data, two inductive biases for time series imputations are proposed and explicitly encoded through two interpretable structures. Among them, **NSG** captures local temporal continuity via the proximity principle, while **PG** models periodic dependencies by emphasizing values at fixed temporal intervals.

- **MG**, a lightweight and model-agnostic module that adaptively learns the importance of NSG and PG based on the characteristics of the specific input are proposed. MG can be seamlessly integrated into diverse deep imputation architectures, enhancing their flexibility and enabling dynamic adaptation to varying temporal dynamics.

- We integrate MG into advanced deep imputation models and evaluate its performance on nine real-world datasets, achieving an average error reduction of 27.39%. The proposed method consistently enhances imputation accuracy and achieves state-of-the-art results, demonstrating its generality, effectiveness, and broad applicability.

## 2 Related Work

### 2.1 Statistical based imputation

Statistical methods are widely used for time series data imputation. A fundamental strategy is using the mean or median value of time series data for imputation [1]. Since most time series data usually do not present sudden changes, researchers [12] propose to utilize $k$ nearest neighbors to impute missing values of the incomplete tuple. In addition, regression models [1] are considered for time series imputation as well, since they can capture intertemporal and intratemporal dependencies within the sequences. Auto Regressive (AR) [21] and ARIMA [31] use the observed values at previous time points to predict missing data in the univariate time series. Furthermore, to obtain more accurate fillings, MICE [2] iteratively creates multiple fillings for obtaining the final imputation.

In general, although statistical methods are explainable and easy to use, they usually rely on strong assumptions and cannot effectively capture complex temporal dependencies in real-world time series.

### 2.2 Deep Learning based imputation

Deep learning models are inherently good at capturing complex and non-linear relationships among data. RNN-based models are more common in earlier works. GRU-D [5], a variant of the gated recurrent unit (GRU), is proposed to address missing data in time series classification tasks. Soon after, BRITS [4] imputes missing values using a bidirectional recurrent dynamical system, without specific assumptions. In the case of M-RNN [45], it imputes missing values based on hidden states derived from bidirectional RNNs. Since the generation ability is naturally suited to the imputation task, generative models are then widely used in filling missing values. E2GAN [20] incorporates a generator based on GRUI within an auto-encoder framework. For spatiotemporal sequence imputation, NAOMI [18] introduces a non-autoregressive model to comprise a bidirectional encoder and a multiresolution decoder. With diffusion models gaining popularity, CSDI [34] emerges, which is a conditional score-based diffusion model designed for time-series imputation. Advancing further, TimesNet [41] expands the examination of temporal variations into the two-dimensional space, while considering the presence of multiple periodicity in time series data. NRTSI [30], an approach for time-series imputation that treats time series as a collection of time-data pairs. SAITS [8] conducts simultaneous reconstruction and imputation by employing a weighted combination of two diagonally-masked self-attention blocks. PSW-I [39] leverages optimal transport with spectral regularization to impute time-series data under temporal and distributional shifts.

Although the aforesaid methods optimize the process of imputing time-series data in various aspects, they do not explicitly encode the characteristics of non-stationarity and periodicity as inductive bias

for time series imputation. This oversight leaves significant room for improvement when dealing with missing data that exhibits complex non-stationarity and periodicity.

## 2.3 Time series characteristics

Time series usually exhibit complex characteristics, with non-stationarity being one of them. Traditional statistical methods generally stabilize data through decomposition or differencing [3]. Adaptive Norm [27] employs a method known as z-score normalization to normalize sampled data segments. DAIN [28] trains a nonlinear neural network to adaptively stabilize the dataset. RevIN [15] performs a normalization and a reverse normalization before and after the model [37], to bring the distributions of various time segments closer together. Periodicity is another important characteristic of time series. To identify the periodicity in complex time series, cyclical or seasonal components are decomposed from the time series. Classical time series decomposition methods, such as X11 [6], iteratively apply moving averages to decompose the time series into multiple components. Fast Fourier Transform (FFT) [23] identifies the potential periodicity by efficiently transforming time series data from the time domain to the frequency domain.

While the existing methods are primarily designed to mitigate non-stationarity or capture periodicity for specific tasks other than imputation, we propose an imputation-oriented inductive bias that jointly models both properties.

## 3 Method

**Problem Formulation**  Given a collection of multivariate time series $\mathbf{X} = \{x_{1:T,1:C}\} \in \mathbb{R}^{T \times C}$ with $T$ timestamps and $C$ channels (attributes). The imputation task is to impute the missing values in $\mathbf{X}$. Formally, an observation mask is defined as $\mathbf{M} = \{m_{1:T,1:C}\} \in \mathbb{R}^{T \times C}$ where $m_{t,c} = 0$ if $x_{t,c}$ is missing, and $m_{t,c} = 1$ if $x_{t,c}$ is observed.

### 3.1 The Impact of Non-stationarity and Periodicity in Time-series Imputation

Time series often exhibit complex dynamics characterized by varying degrees of non-stationarity and periodicity, which motivates us to design inductive biases that adapt to these properties. To design an effective approach that incorporates these properties, we first investigate how non-stationarity and periodicity manifest in real-world time series data.

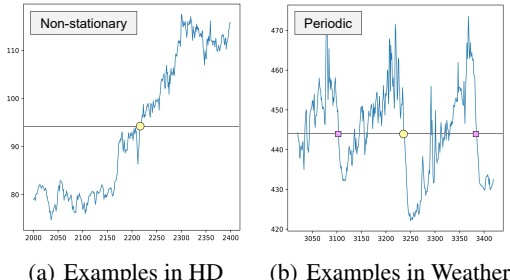

(a) Examples in HD      (b) Examples in Weather

Figure 1: (a) An example in HD Dataset which shows a higher degree of non-stationarity. (b) An example in Weather Dataset which shows a higher degree of periodicity.

Specifically, we adopt the Augmented Dickey-Fuller (ADF) test statistic [10] to quantitatively assess the degree of stationarity, where a higher ADF statistic indicates stronger non-stationarity. As illustrated in Figure 1(a), we present a representative non-stationary time series segment from the HD dataset, with an ADF statistic of 1.958. In such sequences, the trend component is difficult to infer from distant historical observations. This motivates a natural inductive bias: the imputation model should prioritize temporally local information when handling non-stationary data.

In contrast, periodicity offers a more straightforward inductive prior. As shown in Figure 1(b), for a missing value, observations located at regular intervals (i.e., one or more period lengths apart) are more likely to be similar. This property suggests that periodic dependencies can be exploited by assigning greater importance to such temporally aligned points during imputation.

**Overall Framework**  Based on the above intuition, our approach consists of three major steps: (1) Learning Non-Stationary Guidance to inject inductive bias considering non-stationarity into imputation model; (2) Learning Periodic Guidance to inject inductive bias taking advantage of the properties of periodicity into the imputation model; (3) Learning Meta Guidance by learning to weigh Non-Stationary Guidance matrices and Periodic Guidance matrices using a meta weighting network.

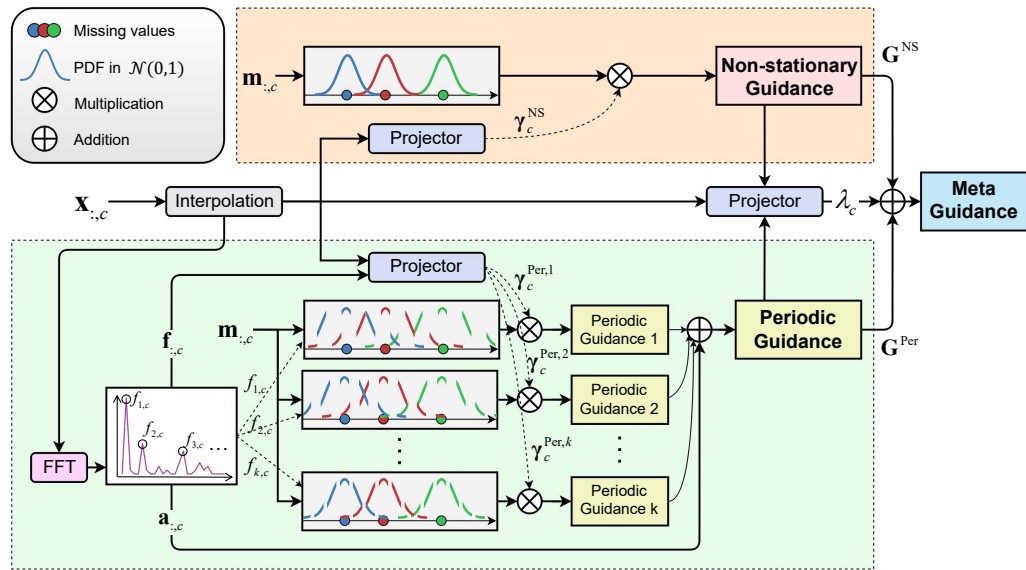

Figure 2: Three steps to learn Meta Guidance: (1) Learning Non-Stationary Guidance, represented in the orange area. (2) Learning Periodic Guidance, represented in the green area. (3) Learning to weigh two guidances considering the properties of input data via a meta weighting network. Appendix H presents visualized examples of NSG and PG constructions.

Meta Guidance can automatically adjust the weight of two types of inductive bias based on the degree of non-stationarity and periodicity in specific time series data through a meta weighting approach.

## 3.2 Learning Non-Stationary Guidance

As aforementioned, Non-Stationary Guidance (NSG) can be designed following the insight that the imputation model should focus more on the values near the missing data rather than the values far from the missing data. Our aim is to learn a Non-Stationary Guidance matrix $\mathbf{G}^{\mathrm{NS}} = \{g^{\mathrm{NS}}_{1:T,1:C}\} \in \mathbb{R}^{T \times C}$. Since channel independence is proven to work well in time-series analysis on both linear models [46], CNNs [47], and transformers [26], we design $\mathbf{G}^{\mathrm{NS}}$ in a channel-independent style, i.e., $g^{\mathrm{NS}}_{t,c}$ is only related to the value of $\mathbf{m}_{:,c} = [m_{1,c}, \cdots, m_{T,c}]$. The core insight in designing NSG is "proximity principle", i.e., values near the missing value can give a guidance on filling the missing value, and the smaller the distance, the stronger the guidance. To achieve this goal, the elements in NSG are defined as follows:

$$g^{\mathrm{NS}}_{t,c} = \sum_{i=-r}^{i=r} \psi(i) \cdot \gamma^{\mathrm{NS}}_c \cdot \mathbb{1}(1 \le i + t \le T) \cdot \mathbb{1}(m_{i+t,c} = 0), \tag{1}$$

where $\psi(i)$ is the probability density of standard normal distribution $\mathcal{N}(0,1)$ when the variable is $i$. $r$ is a hyperparameter to determine the distance within which neighbors can provide guidance for missing values. $\gamma^{\mathrm{NS}}_c$ is a scaling scalar in channel $c$ which can dynamically adjust the strength of guidance and is learned by a two-layer Multi-Layer Perceptron (MLP). The symbol $\mathbb{1}(\cdot)$ denotes the indicator function. The existence of missing data impedes the capability of networks to learn the inherent properties of the data slice, thus we adopt vanilla linear interpolation to the missing data:

$$\mathbf{X}' = \mathrm{Interpolation}(\mathbf{X}, \mathbf{M}), \tag{2}$$

then $\gamma^{\mathrm{NS}}_c$ can be learned via MLP:

$$\log \gamma^{\mathrm{NS}}_c = \mathrm{MLP}\left(\mathbf{x}'_{:,c}\right), \tag{3}$$

where $\mathbf{x}'_{:,c} = \left[x'_{1,c}, x'_{2,c}, \ldots, x'_{T,c}\right]$. The usage of logarithm is to ensure that $\gamma^{\mathrm{NS}}_c$ is non-negative, and channel-wise scaling scalar can make the guidance adapt to different channels with varying degrees of non-stationarity. The architecture of $\mathbf{G}^{\mathrm{NS}}$ encourages imputation algorithms to prioritize local contextual patterns surrounding missing data entries.

## 3.3 Learning Periodic Guidance

In stationary time series, the periodic component typically dominates over the trend component, making it essential to incorporate periodicity-aware inductive bias into the imputation process. As discussed in Section 3.1, values separated by regular intervals—typically one or more period lengths—tend to exhibit higher similarity. Therefore, for time series exhibiting strong periodicity, the imputation model should assign greater attention to observations occurring at periodic offsets from the missing timestamps. To enable this, we first estimate the dominant period lengths for each channel of the input sequence. We adopt Fast Fourier Transform (FFT) to analyze each channel of time series data in the frequency domain,

$$\mathbf{a}_{:,c} = \mathrm{Amp}\left(\mathrm{FFT}\left(\mathbf{x}'_{:,c}\right)\right), \tag{4}$$

where $\mathrm{FFT}(\cdot)$ and $\mathrm{Amp}(\cdot)$ denote the FFT and the calculation of amplitude values, $\mathbf{a}_{:,c}$ indicate the amplitude of each frequency in channel $c$. Then we select the top-$k$ amplitude values and obtain the most significant frequencies:

$$\{f_{1,c}, \cdots, f_{k,c}\} = \underset{f_{*,c} \in \{1, \cdots, [\frac{T}{2}]\}}{\arg \mathrm{Topk}} (\mathbf{a}_{:,c}). \tag{5}$$

We can get the corresponding period lengths as follows:

$$p_{l,c} = \left\lceil \frac{T}{f_{l,c}} \right\rceil, \tag{6}$$

Similar to Non-Stationary Guidance, we design Periodic Guidance (PG) in a channel-independent style. For the period length corresponding to $l$-th significant frequencies, the elements of PG is defined as follows:

$$g_{t,c}^{\mathrm{Per},l} = \sum_{i=-r}^{i=r} \psi(i) \cdot \gamma_c^{\mathrm{Per},l} \cdot \mathbb{1}(1 \le i \cdot p_{l,c} + t \le T) \quad \cdot \mathbb{1}(m_{i \cdot p_{l,c}+t,c} = 0), \tag{7}$$

where $\gamma_c^{\mathrm{Per},l}$ is a scaling scalar in channel $c$ which can dynamically adjust the strength of guidance and is learned by a two-layer Multi-Layer Perceptron (MLP):

$$\log \gamma_c^{\mathrm{Per},l} = \mathrm{MLP}\left(\mathbf{x}'_{:,c}, f_{l,c}\right). \tag{8}$$

The usage of logarithm is to ensure that $\gamma_c^{\mathrm{Per},l}$ is non-negative, and channel-wise scaling scalar can make the guidance adapt to different channels with different degrees of periodicity. The selected frequencies $f_{l,c}$ are also fed into the MLP as extra information.

For each frequency $f_{l,c}$ in channel $c$, we can obtain a PG vector $\mathbf{g}_{:,c}^{\mathrm{Per},l}$ according to Equation 7. The overall PG vector in channel $c$ can be fused depending on how significant each frequency is, which can be measured by the amplitude $a_{l,c}$ of each frequency $f_{l,c}$, as in Equation 4:

$$\mathbf{g}_{:,c}^{\mathrm{Per}} = \sum_{l=1}^{k} \frac{a_{l,c}}{\sum_{l'=1}^{k} a_{l',c}} \mathbf{g}_{:,c}^{\mathrm{Per},l}. \tag{9}$$

## 3.4 Learning Meta Guidance for Imputation

Effectively obtaining conclusive guidance utilizing NSG and PG remains a challenge, particularly when addressing the varied degrees of non-stationarity and periodicity present in different time-series data. To tackle this challenge, we propose Meta Guidance (MG), which automatically learns the weight of NSG and PG in different time-series slices. Similar to the design of the scaling scalars in Equation 3 and Equation 8, we design the weighting scalar $\lambda_c$ in a channel-independent style:

$$\lambda_c = \mathrm{MLP}(\mathbf{x}'_{:,c}, \mathbf{g}_{:,c}^{\mathrm{NS}}, \mathbf{g}_{:,c}^{\mathrm{Per}}), \tag{10}$$

where $\mathbf{g}_{:,c}^* = \left[g_{1,c}^*, g_{2,c}^* \ldots, g_{T,c}^*\right]$ is the channel-wise guidance, the activation function in the final layer of MLP is Sigmoid to ensure that $\lambda_c \in [0, 1]$. Then we can get the channel-wise MG:

$$\mathbf{g}_{:,c}^{\mathrm{Meta}} = \lambda_c \cdot \mathbf{g}_{:,c}^{\mathrm{NS}} + (1 - \lambda_c) \cdot \mathbf{g}_{:,c}^{\mathrm{Per}}. \tag{11}$$

Finally we can get a MG matrix consisting of $c$ channel-wise MG vectors: $\mathbf{G} = \left[\mathbf{g}_{:,1}^{\mathrm{Meta}}, \cdots, \mathbf{g}_{:,c}^{\mathrm{Meta}}\right]^{\mathsf{T}}$.

### 3.5 Injecting Meta Guidance on Transformer

Meta Guidance assigns importance weights to different time points during the imputation of missing values and can be seamlessly integrated into advanced deep learning architectures, such as Transformer-based models. To illustrate the integration process, we use the vanilla Transformer as a representative example. Detailed descriptions of how MG is incorporated into other model architectures are provided in Appendix A.

**Injecting MG into Embedding** In transformer, $\mathbf{X}$ is first embedded into feature representation:

$$\mathbf{E}^{\text{Input}} = \text{Embedding}\left(\mathbf{X}\right) + \mathbf{P}, \qquad (12)$$

where $\mathbf{P}$ is positional encoding. An intuitive idea is to use $\mathbf{G}^{\text{Meta}}$ to provide extra information in representation. $\mathbf{G}^{\text{Meta}}$ is first embedded by an embedding layer:

$$\mathbf{E}^{\text{Meta}} = \text{Embedding}\left(\mathbf{G}^{\text{Meta}} \odot \mathbf{X}\right), \qquad (13)$$

where $\mathbf{G}^{\text{Meta}} \odot \mathbf{X}$ means the Hadamard product of Meta Guidance $\mathbf{G}^{\text{Meta}}$ and input $\mathbf{X}$. The adoption of Hadamard product is because it can achieve the goal of assigning importance weight to each data point $x_{t,c}$. Then we use concatenation to fusion the two representations:

$$\mathbf{E} = \left[\mathbf{E}^{\text{Input}}, \mathbf{E}^{\text{Meta}}\right], \qquad (14)$$

$\mathbf{E}$, instead of $\mathbf{E}^{\text{Input}}$, is then feed forward to the next block.

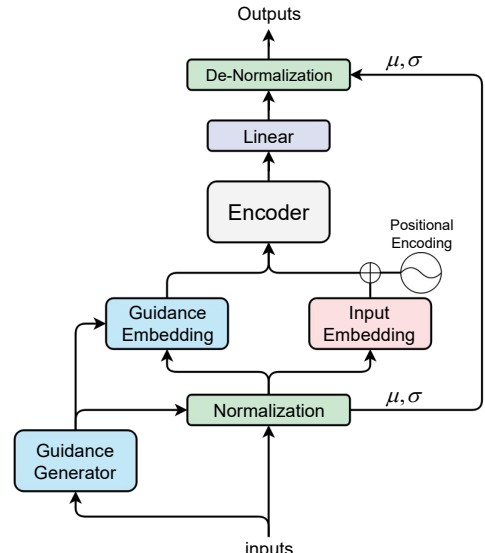

Figure 3: The overall framework of injecting Meta Guidance to the Transformer. Compared to the vanilla Transformer, it includes additional components: the Guidance Generator, the Guidance Embedding module, Normalization and De-normalization modules.

**Injecting MG into Normalization** Normalization techniques are essential for deep learning architectures, and instance normalization [37] is proven effective for time-series analysis. Inspired by that, the advanced normalization method, RevIN [15], normalizes the input to the model with learnable parameters, aligning the distribution of each sequence, and applies an inverse transformation (de-normalization) to the model's output. As aforementioned, Meta Guidance can assign an importance weight to each data point. We apply this idea to modify the learning process of the normalization module. Specifically, we calculated weighted mean and weighted variance using Meta Guidance as the specific weight:

$$\mu_c = \frac{\sum_{t=1}^{T}\left(g_{t,c}^{\text{Meta}} \cdot x_{t,c}\right)}{\sum_{t=1}^{T} g_{t,c}^{\text{Meta}}}, \quad \sigma_c^2 = \frac{\sum_{t=1}^{T}\left(x_{t,c} - \mu_c\right)^2 \cdot g_{t,c}^{\text{Meta}}}{\sum_{i=1}^{T} g_{t,c}^{\text{Meta}}}, \quad \bar{x}_{t,c} = \frac{1}{\sigma_c} \cdot \left(x_{t,c} - \mu_c\right). \quad (15)$$

Also, in the de-normalization module, we inverse transform the output of the model $\mathcal{H}$ using the same mean and variance in each channel $c$:

$$\hat{y}_{t,c} = \sigma_c \cdot \bar{y}_{t,c} + \mu_c, \qquad (16)$$

where $\bar{y}_{t,c}$ is the prediction of the base model, and $\hat{y}_{t,c}$ is the prediction processed by de-normailzation module.

## 4 Experiments

In this section, we incorporate our framework into several advanced deep imputers and compare them with state-of-the-art time series imputation methods. The results indicate that the performance of all methods improves by incorporating our framework. Our codes are available at https://github.com/yjcGitHub0/Meta-Guidance.

Table 1: Summary of datasets, where a larger ADF indicates greater non-stationarity.

| Dataset | HD | Electricity | Traffic | Weather | ETTh1 | ETTh2 | ETTm1 | ETTm2 | TCPC |
|---------|------|-------------|---------|---------|--------|--------|---------|---------|---------|
| $C$ | 5 | 321 | 862 | 21 | 7 | 7 | 7 | 7 | 8 |
| $T$ | 3,020 | 26,304 | 17,544 | 52,696 | 17,420 | 17,420 | 69,680 | 69,680 | 52,416 |
| ADF | 1.958 | -8.444 | -15.021 | -26.681 | -5.908 | -4.136 | -14.984 | -5.663 | -22.199 |

## 4.1 Experimental Setup

**Datasets**  Nine real-world IoT datasets are used in experiments. Table 1 shows the main statistics of each dataset. (1) **HD** is a subset of the DJIA 30 Stock Time Series Dataset [7], which collects daily stock trading information of HD company over 12 years, exhibiting high non-stationarity. (2) **Weather** [40] comprises 21 weather indicators with data recorded every 10 minutes in 2020. (3) **Electricity** [36] collects hourly electricity consumption data of 321 customers from 2012 to 2014. (4) **Traffic** [29] describes road occupancy rates and contains hourly sensor data of San Francisco freeways from 2015 to 2016. (5) **ETT** [48] consists of two hourly datasets (ETTh) and two 15-minute datasets (ETTm). Each of them includes load characteristics of seven types of oil and power transformers from July 2016 to July 2018. (6) **TCPC** (Tetuan City power consumption) [33], which is collected in a room to estimate occupancy, featuring temperature, light, sound, $CO_2$, and PIR sensors.

**Baselines**  We integrate our framework into **Transformer** [38], **CSDI** [34], **TimesNet** [41], **SAITS** [8], and **iTransformer** [17], since they yield the state-of-the-art performance in most cases and the process of integrating Meta Guidance into them is fundamentally similar. Regarding other methods, such as **M-RNN** [45], **GAIN** [44], **BRITS** [4], **TIDER** [16], **ImputeFormer** [25], **ModernTCN** [19], and **PSW-I** [39], we utilize them as baselines for comparison. Due to space constraints, we report results for a subset of advanced baselines and representative datasets in the main paper. The full experimental results can be found in Table 10 and 11 of Appendix G.

**Implementation Details**  We follow the configuration and parameters of TimesNet [41], using the same dataset partitioning method, missing pattern, and imputation window length. Specifically, for methods requiring training, testing, and validation sets, we split them in a 7:2:1 ratio; otherwise, we use an 8:2 ratio for training and testing. Missing values are injected using the Missing Completely At Random (MCAR) mechanism, which is adopted in most existing studies. Other missing mechanisms are considered in Appendix D. For methods requiring an imputation window, we set the window length to 96. Transformer-based methods use a consistent set of parameters based on the dataset, matching the settings in TimesNet. Experiments were conducted on a system with 8 NVIDIA RTX 4090 GPUs (24GB VRAM each), a 128-core AMD EPYC 7513 CPU, and 503GB RAM. All reported results are averaged over three independent runs. The standard deviations of the full experiments are provided in Appendix C.

## 4.2 Comparative Experiments

We investigate the imputation performance of different methods on real datasets with varying missing rates, and the corresponding results are presented in Table 2. We observe that the optimal methods across all datasets are those incorporating our framework. For instance, regarding the MAE metric, the Meta Guidance framework leads to an average improvement of 43.76%, 26.49%, 17.91%, 9.88%, 25.82%, 34.61% for the methods on HD, Weather, Electricity, Traffic, ETT and TCPC datasets, respectively. Such results demonstrate that Meta Guidance is an effective lightweight framework applicable to a wide range of deep learning imputation models, significantly enhancing their imputation performance.

We observe that traditional imputation methods struggle on highly non-stationary datasets such as HD. In contrast, methods augmented with our MG framework exhibit notably larger performance gains on non-stationary datasets compared to stationary ones. This is attributed to the fact that non-stationary time series exhibit evolving statistical properties and shifting joint distributions, which can mislead data-driven models. Notably, both our method and other recent advances substantially outperform earlier baselines on the HD dataset, demonstrating MG's effectiveness in mitigating non-stationarity. Moreover, MG also yields consistent improvements on stationary datasets with strong periodicity,

Table 2: Imputation results over real datasets with different missing rates. The notation "+MG" signifies that the method incorporates our MG framework. "Promotion" denotes the average reduction in MAE for the corresponding dataset. We evaluate our approach across 9 datasets and compare it with 12 baseline methods. See Table 10 and 11 for full results.

| dataset | | HD | | Weather | | Electricity | | Traffic | | ETTm1 | | TCPC | |
|---|---|---|---|---|---|---|---|---|---|---|---|---|---|
| metric | | MAE | RMSE | MAE | RMSE | MAE | RMSE | MAE | RMSE | MAE | RMSE | MAE | RMSE |
| Transformer | 10% | 0.112 | 0.188 | 0.088 | 0.182 | 0.276 | 0.379 | 0.233 | 0.441 | 0.177 | 0.257 | 0.177 | 0.248 |
| | 25% | 0.119 | 0.234 | 0.095 | 0.196 | 0.284 | 0.394 | 0.231 | 0.453 | 0.213 | 0.308 | 0.233 | 0.328 |
| | 40% | 0.134 | 0.247 | 0.101 | 0.204 | 0.291 | 0.407 | 0.233 | 0.466 | 0.238 | 0.344 | 0.276 | 0.382 |
| Transformer +MG | 10% | **0.046** | **0.082** | **0.049** | **0.159** | **0.180** | **0.261** | **0.198** | **0.388** | **0.108** | **0.169** | **0.069** | **0.118** |
| | 25% | **0.050** | **0.095** | **0.052** | **0.168** | **0.191** | **0.276** | **0.203** | **0.397** | **0.134** | **0.207** | **0.086** | **0.145** |
| | 40% | **0.057** | **0.105** | **0.058** | **0.176** | **0.200** | **0.289** | **0.217** | **0.422** | **0.153** | **0.235** | **0.103** | **0.169** |
| CSDI | 10% | 0.096 | 0.163 | 0.022 | 0.151 | 0.115 | 0.188 | 0.126 | 0.356 | 0.069 | 0.124 | 0.019 | 0.057 |
| | 25% | 0.166 | 0.285 | 0.024 | 0.159 | 0.119 | 0.195 | 0.113 | 0.333 | 0.076 | 0.136 | 0.021 | 0.068 |
| | 40% | 0.151 | 0.265 | 0.027 | 0.172 | 0.126 | 0.205 | 0.123 | 0.358 | 0.085 | 0.151 | 0.024 | 0.074 |
| CSDI +MG | 10% | **0.040** | **0.107** | **0.019** | **0.127** | **0.102** | **0.163** | **0.104** | **0.263** | **0.065** | **0.110** | **0.018** | **0.054** |
| | 25% | **0.041** | **0.107** | **0.022** | **0.139** | **0.113** | **0.180** | **0.097** | **0.264** | **0.073** | **0.126** | **0.021** | **0.062** |
| | 40% | **0.042** | **0.103** | **0.024** | **0.152** | **0.122** | **0.195** | **0.110** | **0.301** | **0.082** | **0.142** | **0.023** | **0.068** |
| TimesNet | 10% | 0.050 | 0.085 | 0.048 | 0.159 | 0.201 | 0.293 | 0.237 | 0.474 | 0.111 | 0.172 | 0.059 | 0.106 |
| | 25% | 0.055 | 0.096 | 0.055 | 0.170 | 0.208 | 0.302 | 0.243 | 0.477 | 0.128 | 0.200 | 0.076 | 0.131 |
| | 40% | 0.058 | 0.105 | **0.063** | **0.180** | 0.216 | 0.312 | 0.253 | 0.489 | 0.145 | 0.225 | 0.092 | 0.154 |
| TimesNet +MG | 10% | 0.050 | 0.085 | **0.046** | **0.158** | **0.194** | **0.280** | **0.231** | **0.463** | **0.095** | **0.150** | **0.045** | **0.088** |
| | 25% | **0.052** | **0.094** | **0.054** | **0.169** | **0.201** | **0.292** | **0.238** | **0.471** | **0.111** | **0.176** | **0.059** | **0.111** |
| | 40% | **0.057** | **0.103** | 0.064 | 0.182 | **0.209** | **0.303** | **0.246** | **0.482** | **0.126** | **0.200** | **0.070** | **0.128** |
| SAITS | 10% | 0.078 | 0.166 | 0.044 | 0.166 | 0.314 | 0.448 | 0.228 | 0.493 | 0.196 | 0.314 | 0.060 | 0.109 |
| | 25% | 0.095 | 0.202 | 0.052 | 0.179 | 0.321 | 0.455 | 0.238 | 0.510 | 0.242 | 0.411 | 0.073 | 0.133 |
| | 40% | 0.113 | 0.214 | 0.080 | 0.202 | 0.344 | 0.482 | 0.274 | 0.548 | 0.362 | 0.568 | 0.096 | 0.166 |
| SAITS +MG | 10% | **0.037** | **0.081** | **0.032** | **0.161** | **0.197** | **0.286** | **0.194** | **0.401** | **0.105** | **0.164** | **0.032** | **0.069** |
| | 25% | **0.044** | **0.097** | **0.037** | **0.177** | **0.209** | **0.301** | **0.209** | **0.421** | **0.115** | **0.186** | **0.040** | **0.093** |
| | 40% | **0.055** | **0.111** | **0.046** | **0.182** | **0.234** | **0.332** | **0.244** | **0.462** | **0.148** | **0.234** | **0.057** | **0.121** |
| iTransformer | 10% | 0.107 | 0.153 | 0.081 | 0.193 | 0.184 | 0.262 | 0.198 | 0.390 | 0.155 | 0.236 | 0.087 | 0.136 |
| | 25% | 0.114 | 0.190 | 0.103 | 0.214 | 0.213 | 0.298 | 0.230 | 0.433 | 0.181 | 0.269 | 0.111 | 0.164 |
| | 40% | 0.131 | 0.226 | 0.122 | 0.234 | 0.237 | 0.329 | 0.257 | 0.474 | 0.207 | 0.303 | 0.122 | 0.181 |
| iTransformer +MG | 10% | **0.075** | **0.109** | **0.055** | **0.176** | **0.166** | **0.241** | **0.185** | **0.375** | **0.128** | **0.205** | **0.052** | **0.103** |
| | 25% | **0.071** | **0.110** | **0.055** | **0.185** | **0.186** | **0.267** | **0.208** | **0.411** | **0.142** | **0.226** | **0.064** | **0.119** |
| | 40% | **0.073** | **0.118** | **0.060** | **0.192** | **0.205** | **0.293** | **0.231** | **0.448** | **0.156** | **0.244** | **0.076** | **0.138** |
| Promotion | | ↑ 43.76% | | ↑ 26.49% | | ↑ 17.91% | | ↑ 9.88% | | ↑ 25.82% | | ↑ 34.61% | |

such as Electricity, by guiding models to better capture periodic patterns. We further present the performance of MG under different missing patterns in Appendix D.

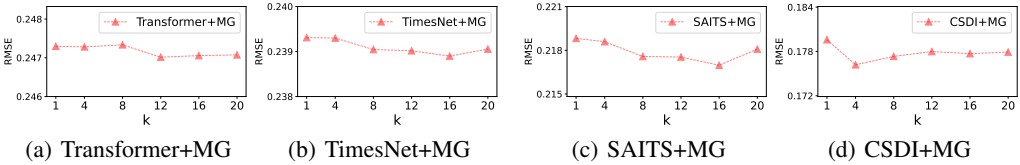

(a) Transformer+MG  (b) TimesNet+MG  (c) SAITS+MG  (d) CSDI+MG

Figure 4: Varying the hyperparameter value of $k$ over ETTh1 dataset with 10% missing values.

### 4.3 Analysis

**Hyperparameter $k$** The hyperparameter $k$ controls the maximum number of frequencies selected during period extraction. We explore the impact of different $k$ values on model performance. As shown in Figure 4, both too small and too large values of $k$ result in decreased imputation performance. A small $k$ fails to capture enough frequency information, leading to ineffective Periodic Guidance. Conversely, a large $k$ introduces noise frequencies that do not aid imputation, potentially distracting the model. The optimal value of $k$ varies across models, depending on their ability to perceive frequency and period.

**Hyperparameter $r$** The hyperparameter $r$ controls the neighborhood range for providing guidance to missing values. As shown in Figure 5, model performance becomes stable when $r \geq 3$, since the Guidance Generator applies a standard normal distribution centered on missing values. Beyond this range, additional neighbors contribute negligible weights, making larger $r$ values redundant.

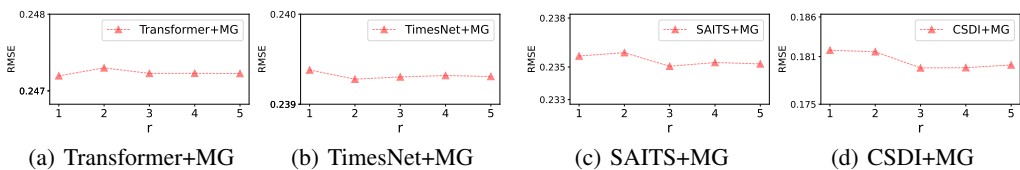

| (a) Transformer+MG | (b) TimesNet+MG | (c) SAITS+MG | (d) CSDI+MG |

Figure 5: Varying the hyperparameter value of $r$ over ETTh1 dataset with 10% missing values.

Conversely, too small an $r$ truncates the distribution, distorting the Guidance and causing unstable results. To ensure both stability and efficiency, we set $r = 3$ by default.

Table 3: Imputation results obtained by applying different methods to Transformer.

| $\mathbf{G}^{\mathrm{NS}}$ | $\mathbf{G}^{\mathrm{Per}}$ | $\mathbf{G}^{\mathrm{Meta}}$ | | HD | | | Electricity | | | Weather | | | TCPC | | |
|---|---|---|---|---|---|---|---|---|---|---|---|---|---|---|---|
| | | | | 10% | 25% | 40% | 10% | 25% | 40% | 10% | 25% | 40% | 10% | 25% | 40% |
| - | - | - | MAE | 0.112 | 0.119 | 0.134 | 0.276 | 0.284 | 0.291 | 0.088 | 0.095 | 0.101 | 0.177 | 0.233 | 0.276 |
| | | | RMSE | 0.188 | 0.234 | 0.247 | 0.379 | 0.394 | 0.407 | 0.182 | 0.196 | 0.204 | 0.248 | 0.328 | 0.382 |
| ✓ | - | - | MAE | 0.047 | 0.052 | 0.058 | 0.182 | 0.194 | 0.202 | 0.053 | 0.056 | 0.064 | 0.076 | 0.112 | 0.115 |
| | | | RMSE | 0.083 | 0.096 | 0.107 | 0.262 | 0.279 | 0.292 | 0.161 | 0.172 | 0.182 | 0.131 | 0.176 | 0.178 |
| - | ✓ | - | MAE | 0.051 | 0.056 | 0.062 | 0.186 | 0.208 | 0.208 | 0.051 | 0.055 | 0.061 | 0.072 | 0.093 | 0.104 |
| | | | RMSE | 0.088 | 0.099 | 0.112 | 0.267 | 0.288 | 0.298 | 0.160 | 0.174 | 0.179 | 0.127 | 0.159 | 0.175 |
| ✓ | ✓ | ✓ | MAE | **0.046** | **0.050** | **0.057** | **0.180** | **0.191** | **0.200** | **0.049** | **0.052** | **0.058** | **0.069** | **0.086** | **0.103** |
| | | | RMSE | **0.082** | **0.095** | **0.105** | **0.261** | **0.276** | **0.289** | **0.159** | **0.168** | **0.176** | **0.118** | **0.148** | **0.169** |

**Ratio $\lambda$** To statistically validate the behavior patterns of Meta Guidance across datasets with varying characteristics, we compute the average $\lambda$ for Transformer and SAITS across different datasets, which represents the proportion of Non-Stationary Guidance within Meta Guidance. The results, shown in Figure 6, indicate that datasets with higher non-stationarity exhibit a higher $\lambda$, reflecting the dominance of Non-Stationary Guidance. Conversely, stable datasets with strong periodicity show a lower $\lambda$. This demonstrates that our method can automatically adapt to different dataset characteristics, making Meta Guidance broadly applicable.

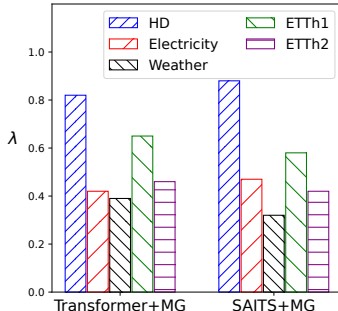

Figure 6: Average $\lambda$ values.

**Ablation Study** To further investigate the impact of individual components of our framework, we conduct ablation studies on the Transformer. The experiment results are presented in Table 3. As shown, the variant without any component represents the vanilla Transformer. The variant with only $\mathbf{G}^{\mathrm{NS}}$ or $\mathbf{G}^{\mathrm{Per}}$ indicates the use of the Meta Guidance framework, but only Non-Stationary Guidance or Periodic Guidance is used in calculating Meta Guidance. Having $\mathbf{G}^{\mathrm{Meta}}$ signifies the use of the complete Meta Guidance framework. We observe that, compared to the vanilla Transformer, both Non-Stationary Guidance and Periodic Guidance can boost the performance of the original model. This indicates that inductive bias for non-stationarity and periodicity can boost time series imputation. Our method effectively addresses both aspects, reducing the adverse effects of complex temporal properties on imputation.

**Time Complexity** We conduct a time consumption evaluation on the HD dataset. The training time for one epoch of the vanilla Transformer and the Transformer with Meta Guidance are 2.6 seconds and 2.9 seconds, respectively. The extra time overhead is acceptable. Additional analysis of time complexity is provided in Appendix F.

## 5 Conclusion

In this paper, we propose two inductive biases for time-series imputation: Non-Stationary Guidance for non-stationary series and Periodic Guidance for periodic series. Meta Guidance is derived by learning to weigh these two guidance matrices based on the input data's characteristics. As a lightweight framework, Meta Guidance can be integrated into deep imputation models to significantly improve their performance. Experiments on nine real-world datasets demonstrate its versatility and effectiveness.

## Acknowledgements

This work was supported by the National Natural Science Foundation of China (62306085, 62302241, 62476071, U23B2055, U24A20328), Shenzhen College Stability Support Plan (GXWD20231130151329002), CCF-ALIMAMA TECH Kangaroo Fund (CCF-ALIMAMA OF 2025001), Guangdong Basic and Applied Basic Research Foundation (2025A1515011732) and Shenzhen Science and Technology Program (KQTD20240729102154066, ZDSYS20230626091203008).

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

# A  Methods for applying Meta Guidance to other models

## A.1  TimesNet+MG

The overall structure of TimesNet is very similar to that of the Transformer, making it easy to apply the Transformer+MG (Meta Guidance) method without needing any changes. The only aspect that needs attention is TimesNet's proprietary Normalization/De-normalization module. To align with the Meta Guidance approach, we have to switch out this built-in module for the Meta Guidance version.

## A.2  iTransformer+MG

The architecture of iTransformer is also similar to that of the Transformer. The only notable difference is a dimension permutation operation applied during the input embedding stage. To seamlessly integrate MG into iTransformer, it is sufficient to apply the same dimension permutation to the MG sequence.

## A.3  SAITS+MG

SAITS is also based on the Transformer architecture, yet it distinguishes itself by incorporating a dual-Transformer configuration. Specifically, the primary Transformer is specifically engineered to enhance the imputation of missing data, whereas the secondary Transformer is optimized to more accurately fit the observed data. Our Meta Guidance framework is meticulously tailored to augment the imputation process and is, therefore, exclusively integrated into the first Transformer, preserving the original functionality of the second Transformer intact. The integration methodology employed parallels that of the Transformer+MG approach.

## A.4  CSDI+MG

Distinct from preceding models, CSDI leverages a conditional diffusion model framework, presenting a structural deviation from the Transformer architecture. This foundational difference notwithstanding, the deep learning underpinnings of the diffusion model permit a streamlined adaptation of the Transformer+MG methodology for CSDI+MG implementation, necessitating minimal adjustments:

1) Throughout the training phase, CSDI employs the actual values of missing data points to generate a noisy target. It is imperative, therefore, within the Guidance Generator and the normalization process, to preserve the original values of missing data in x rather than defaulting them to zero. For these missing values, Meta Guidance is assigned a value of 1. Subsequent to the generation of the noisy target, these missing values are nullified (set to 0) to preclude the model from accessing extra information pertaining to the missing data.

2) In addition to the conditional observation, CSDI introduces an auxiliary condition termed "Side information", encompassing Time Embedding, Feature Embedding, and the conditional mask. The matrix produced by Guidance Embedding is integrated at this juncture, amalgamated with the initial Side information, and subsequently employed as a conditional input to the diffusion model.

3) The training loss for the diffusion model is computed based on the variance from random noise, diverging from the Transformer's approach of contrasting the imputed missing values against their actual counterparts. This distinction necessitates the exclusion of the De-normalization transformation from the training output. The Normalization module is exclusively retained during the training process. Conversely, in the evaluation and testing phases, when the diffusion model generates the imputed sequence, the comprehensive Normalization/De-normalization module is reinstated.

# B  Additional Experiments on the Hyperparameters $k$ and $r$

To further validate the stability and robustness of Meta Guidance (MG), we conduct an extended investigation into its sensitivity with respect to key hyperparameters under different missing-rate conditions. Specifically, we focus on two important hyperparameters, $k$ and $r$. These hyperparameters jointly influence the receptive field and the inductive bias of MG. Evaluating their effects helps us understand how the model behaves when these configurations vary.

We systematically examine the performance of MG-enhanced models across a wide range of $k$ and $r$ values. As shown in Tables 4–7, the results reveal that the MG module consistently achieves stable performance across different settings. The variance among configurations is marginal, indicating that our method is largely insensitive to hyperparameter choices and can be applied without extensive tuning. This property further highlights MG's practical robustness and adaptability when deployed in diverse real-world time-series applications.

Table 4: Varying the hyperparameter value of $k$ over ETTh1 dataset with 25% missing values.

| 25% missing rate | k=1 | k=4 | k=8 | k=12 | k=16 | k=20 |
|---|---|---|---|---|---|---|
| Transformer+MG | 0.299 | 0.299 | 0.307 | 0.319 | 0.299 | 0.299 |
| TimesNet+MG | 0.288 | 0.288 | 0.288 | 0.288 | 0.288 | 0.288 |
| SAITS+MG | 0.282 | 0.280 | 0.280 | 0.279 | 0.279 | 0.280 |
| CSDI+MG | 0.213 | 0.210 | 0.212 | 0.213 | 0.212 | 0.215 |

Table 5: Varying the hyperparameter value of $k$ over ETTh1 dataset with 40% missing values.

| 40% missing rate | k=1 | k=4 | k=8 | k=12 | k=16 | k=20 |
|---|---|---|---|---|---|---|
| Transformer+MG | 0.347 | 0.347 | 0.346 | 0.346 | 0.346 | 0.346 |
| TimesNet+MG | 0.319 | 0.319 | 0.319 | 0.319 | 0.319 | 0.320 |
| SAITS+MG | 0.355 | 0.358 | 0.358 | 0.358 | 0.356 | 0.358 |
| CSDI+MG | 0.266 | 0.269 | 0.268 | 0.265 | 0.264 | 0.263 |

Table 6: Varying the hyperparameter value of $r$ over ETTh1 dataset with 25% missing values.

| 25% missing rate | r=1 | r=2 | r=3 | r=4 | r=5 |
|---|---|---|---|---|---|
| Transformer+MG | 0.338 | 0.299 | 0.301 | 0.317 | 0.307 |
| TimesNet+MG | 0.288 | 0.288 | 0.288 | 0.288 | 0.288 |
| SAITS+MG | 0.279 | 0.279 | 0.280 | 0.283 | 0.280 |
| CSDI+MG | 0.209 | 0.215 | 0.207 | 0.212 | 0.215 |

Table 7: Varying the hyperparameter value of $r$ over ETTh1 dataset with 40% missing values.

| 40% missing rate | r=1 | r=2 | r=3 | r=4 | r=5 |
|---|---|---|---|---|---|
| Transformer+MG | 0.348 | 0.347 | 0.346 | 0.346 | 0.346 |
| TimesNet+MG | 0.321 | 0.320 | 0.319 | 0.319 | 0.319 |
| SAITS+MG | 0.356 | 0.356 | 0.357 | 0.356 | 0.356 |
| CSDI+MG | 0.267 | 0.265 | 0.266 | 0.266 | 0.266 |

## C  Details on the standard deviation

Standard deviation measures how much data deviates from the mean, reflecting the variability or consistency of a dataset—larger values indicate greater dispersion, while smaller values suggest tighter clustering around the mean. In our experiments, all results are averaged over three independent runs. Table 8 reports the corresponding standard deviations.

Table 8: Details on the standard deviation of methods that incorporate the Meta Guidance framework.

| stdev | HD | Weather | Electricity | Traffic | ETTh1 | ETTh2 | ETTm1 | ETTm2 | TCPC |
|---|---|---|---|---|---|---|---|---|---|
| M-RNN | 0.1794 | 0.0164 | 0.0011 | 0.0021 | 0.0351 | 0.0555 | 0.0357 | 0.0564 | 0.0263 |
| GAIN | 0.0249 | 0.0764 | 0.0026 | 0.0016 | 0.0995 | 0.0663 | 0.0355 | 0.0399 | 0.0662 |
| BRITS | 0.0486 | 0.0022 | 0.0014 | 0.0019 | 0.0085 | 0.0080 | 0.0108 | 0.0060 | 0.0030 |
| TIDER | 0.0170 | 0.0789 | 0.0241 | 0.0290 | 0.0110 | 0.0095 | 0.0207 | 0.0136 | 0.0074 |
| Imputeformer | 0.0170 | 0.0789 | 0.0241 | 0.0290 | 0.0110 | 0.0095 | 0.0207 | 0.0136 | 0.0136 |
| ModernTCN | 0.0170 | 0.0789 | 0.0241 | 0.0290 | 0.0110 | 0.0095 | 0.0207 | 0.0136 | 0.0136 |
| PSW-I | 0.0079 | 0.0208 | 0.0002 | 0.0007 | 0.0015 | 0.0008 | 0.0005 | 0.0003 | 0.0003 |
| Transformer | 0.0479 | 0.0040 | 0.0009 | 0.0014 | 0.0031 | 0.0032 | 0.0033 | 0.0031 | 0.0060 |
| SAITS | 0.1325 | 0.0028 | 0.0019 | 0.0029 | 0.0240 | 0.0155 | 0.0093 | 0.0164 | 0.0028 |
| TimesNet | 0.0021 | 0.0023 | 0.0013 | 0.0013 | 0.0050 | 0.0013 | 0.0012 | 0.0006 | 0.0014 |
| iTransformer | 0.0093 | 0.0022 | 0.0005 | 0.0006 | 0.0057 | 0.0024 | 0.0013 | 0.0005 | 0.0023 |
| CSDI | 0.0038 | 0.0017 | 0.0015 | 0.0018 | 0.0021 | 0.0018 | 0.0023 | 0.0011 | 0.0016 |
| Transformer+MG | 0.0022 | 0.0018 | 0.0004 | 0.0033 | 0.0050 | 0.0012 | 0.0010 | 0.0007 | 0.0010 |
| SAITS+MG | 0.0023 | 0.0022 | 0.0007 | 0.0021 | 0.0059 | 0.0018 | 0.0044 | 0.0006 | 0.0007 |
| TimesNet+MG | 0.0019 | 0.0027 | 0.0027 | 0.0024 | 0.0038 | 0.0012 | 0.0010 | 0.0006 | 0.0017 |
| iTransformer+MG | 0.0020 | 0.0022 | 0.0004 | 0.0007 | 0.0058 | 0.0010 | 0.0007 | 0.0002 | 0.0014 |
| CSDI+MG | 0.0025 | 0.0010 | 0.0008 | 0.0016 | 0.0022 | 0.0008 | 0.0017 | 0.0005 | 0.0012 |

## D  Missing Pattern Analysis

We consider another two missing data injection mechanisms, i.e., Missing At Random (MAR) [42] and Missing Not At Random (MNAR) [35], following the existing study [22]. Figure 7 shows the experimental results returned by various methods with different missing mechanisms. We observe that the performance of various methods fluctuates slightly under different missing mechanisms, but the methods incorporating Meta Guidance still achieve significant performance improvements. This demonstrates the applicability of Meta Guidance across various types of missing data.

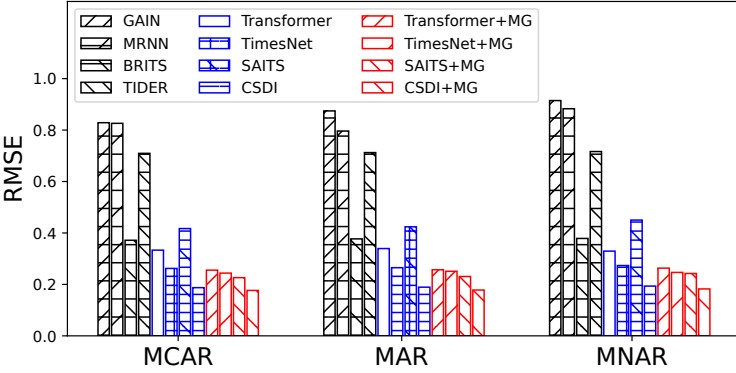

Figure 7: Varying the missing mechanism over ETTh1 dataset with 10% missing values.

# E Showcase

To highlight the effect of Meta Guidance (MG), we present imputation showcases on the ETTm1 dataset with 40% missing data (Figure 8). The vanilla Transformer suffers from short-term periodic fluctuations that deviate from the ground truth, whereas incorporating MG effectively suppresses these artifacts and yields more accurate results. Even strong models like TimesNet benefit from MG, which enhances performance by guiding attention toward informative neighboring values and mitigating unrealistic deviations during imputation.

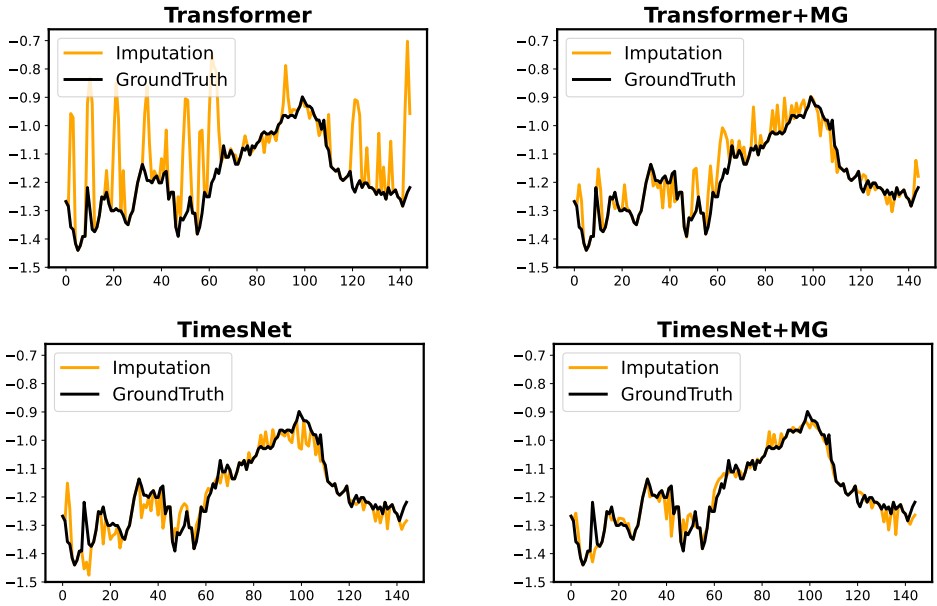

Figure 8: Visualization of ETTm1 imputation results given by models under the 40% mask ratio setting. The black lines stand for the ground truth and the orange lines stand for predicted values.

# F Time Complexity Analysis

Since the construction of NSG and PG follows a linear process, the overall time complexity of MG can be expressed as $\mathrm{O}(Tr) + \mathrm{O}(Td)$, where $T$ represents the dataset length, $r$ is an MG hyperparameter (as defined in Equation 3 of Section 3.1) typically set to around 3, and $d$ denotes the dimensionality of the MLP hidden layer, set to 12 in our experiments. Here, $\mathrm{O}(Tr)$ accounts for the complexity of constructing the raw NSG and PG sequences, while $\mathrm{O}(Td)$ corresponds to the complexity of the Projector's linear layer. Given that both $r$ and $d$ are fixed hyperparameters, the additional computational complexity introduced by MG simplifies to $\mathrm{O}(T)$, making its overhead negligible.

To further validate the computational efficiency of MG, we conducted additional experiments using the SOTA methods (CSDI vs. CSDI+MG) on datasets with more time steps and channels, and the Table 9 shows the training time per iteration across datasets. The results show that our method introduces even less time overhead on larger datasets, approximately 3%, which is acceptable.

Table 9: Runtime per iteration (in seconds) across different datasets.

| time cost (s) | Weather | ETTh1 | Electricity | Traffic |
|---|---|---|---|---|
| CSDI | 0.0363 | 0.0261 | 0.7561 | 1.4512 |
| CSDI+MG | 0.0397 (+9.3%) | 0.0288 (+10.1%) | 0.7783 (+2.9%) | 1.4820 (+2.1%) |

# G   Full experimental results

In this section, we present the full experimental results, which include all baseline methods. As shown in Table 10 and 11, the methods incorporating Meta Guidance outperform all baselines. In particular, CSDI, due to its excellent intrinsic qualities, demonstrates superior performance across most datasets after integrating our framework. Compared to the original methods, the performance of those incorporating Meta Guidance shows significant improvement. Overall, methods incorporating Meta Guidance exhibit a marked performance boost over the original approaches.

Table 10: MAE imputation results on real-world datasets under varying missing rates. The notation "+MG" signifies that the method incorporates our Meta Guidance framework.

| | | HD | Weather | Electricity | Traffic | ETTh1 | ETTh2 | ETTm1 | ETTm2 | TCPC |
|---|---|---|---|---|---|---|---|---|---|---|
| M-RNN | 10% | 0.727 | 0.291 | 0.355 | 0.286 | 0.627 | 0.621 | 0.639 | 0.584 | 0.712 |
| | 25% | 0.831 | 0.341 | 0.368 | 0.302 | 0.672 | 0.638 | 0.672 | 0.616 | 0.722 |
| | 40% | 0.904 | 0.382 | 0.381 | 0.321 | 0.702 | 0.659 | 0.702 | 0.643 | 0.742 |
| GAIN | 10% | 0.209 | 0.268 | 0.257 | 0.180 | 0.596 | 0.610 | 0.558 | 0.598 | 0.701 |
| | 25% | 0.222 | 0.303 | 0.257 | 0.189 | 0.567 | 0.600 | 0.593 | 0.610 | 0.689 |
| | 40% | 0.239 | 0.310 | 0.261 | 0.196 | 0.680 | 0.652 | 0.615 | 0.614 | 0.775 |
| BRITS | 10% | 0.196 | 0.045 | 0.249 | 0.216 | 0.249 | 0.147 | 0.109 | 0.090 | 0.067 |
| | 25% | 0.232 | 0.059 | 0.267 | 0.226 | 0.291 | 0.171 | 0.147 | 0.119 | 0.107 |
| | 40% | 0.279 | 0.077 | 0.292 | 0.244 | 0.339 | 0.202 | 0.183 | 0.152 | 0.164 |
| TIDER | 10% | 0.185 | 0.368 | 0.568 | 0.523 | 0.471 | 0.472 | 0.548 | 0.541 | 0.634 |
| | 25% | 0.224 | 0.378 | 0.575 | 0.539 | 0.527 | 0.518 | 0.593 | 0.576 | 0.640 |
| | 40% | 0.331 | 0.401 | 0.558 | 0.557 | 0.621 | 0.590 | 0.635 | 0.611 | 0.679 |
| Imputeformer | 10% | 0.748 | 0.086 | 0.408 | 0.373 | 0.437 | 0.209 | 0.259 | 0.125 | 0.146 |
| | 25% | 0.829 | 0.090 | 0.424 | 0.368 | 0.455 | 0.232 | 0.281 | 0.136 | 0.155 |
| | 40% | 0.904 | 0.103 | 0.459 | 0.419 | 0.495 | 0.280 | 0.318 | 0.158 | 0.173 |
| ModernTCN | 10% | 0.080 | 0.049 | 0.169 | 0.180 | 0.147 | 0.093 | 0.091 | 0.053 | 0.034 |
| | 25% | 0.076 | 0.062 | 0.192 | 0.206 | 0.161 | 0.097 | 0.099 | 0.057 | 0.035 |
| | 40% | 0.079 | 0.060 | 0.203 | 0.227 | 0.181 | 0.103 | 0.109 | 0.063 | 0.041 |
| PSW-I | 10% | 0.074 | 0.056 | 0.192 | 0.266 | 0.195 | 0.125 | 0.108 | 0.074 | 0.034 |
| | 25% | 0.077 | 0.059 | 0.213 | 0.260 | 0.206 | 0.130 | 0.113 | 0.078 | 0.039 |
| | 40% | 0.084 | 0.063 | 0.243 | 0.285 | 0.225 | 0.140 | 0.121 | 0.083 | 0.046 |
| Transformer | 10% | 0.112 | 0.088 | 0.276 | 0.233 | 0.236 | 0.179 | 0.177 | 0.150 | 0.177 |
| | 25% | 0.119 | 0.095 | 0.284 | 0.231 | 0.276 | 0.203 | 0.213 | 0.166 | 0.233 |
| | 40% | 0.134 | 0.101 | 0.291 | 0.233 | 0.306 | 0.225 | 0.238 | 0.181 | 0.276 |
| SAITS | 10% | 0.078 | 0.044 | 0.314 | 0.228 | 0.273 | 0.222 | 0.196 | 0.129 | 0.060 |
| | 25% | 0.095 | 0.052 | 0.321 | 0.238 | 0.333 | 0.258 | 0.242 | 0.161 | 0.073 |
| | 40% | 0.113 | 0.080 | 0.344 | 0.274 | 0.415 | 0.333 | 0.362 | 0.253 | 0.096 |
| TimesNet | 10% | 0.050 | 0.048 | 0.201 | 0.237 | 0.174 | 0.091 | 0.111 | 0.059 | 0.059 |
| | 25% | 0.055 | 0.055 | 0.208 | 0.243 | 0.211 | 0.107 | 0.128 | 0.070 | 0.076 |
| | 40% | 0.058 | 0.063 | 0.216 | 0.253 | 0.236 | 0.122 | 0.145 | 0.078 | 0.092 |
| iTransformer | 10% | 0.147 | 0.081 | 0.184 | 0.198 | 0.221 | 0.142 | 0.155 | 0.101 | 0.087 |
| | 25% | 0.234 | 0.103 | 0.213 | 0.230 | 0.258 | 0.165 | 0.181 | 0.124 | 0.111 |
| | 40% | 0.340 | 0.122 | 0.237 | 0.257 | 0.291 | 0.184 | 0.207 | 0.144 | 0.122 |
| CSDI | 10% | 0.096 | 0.022 | 0.115 | 0.126 | 0.103 | 0.049 | 0.069 | 0.031 | 0.019 |
| | 25% | 0.166 | 0.024 | 0.119 | 0.113 | 0.118 | 0.058 | 0.076 | 0.036 | 0.021 |
| | 40% | 0.151 | 0.027 | 0.126 | 0.123 | 0.143 | 0.068 | 0.085 | 0.041 | 0.024 |
| Transformer +MG | 10% | 0.046 | 0.049 | 0.180 | 0.198 | 0.169 | 0.089 | 0.108 | 0.058 | 0.069 |
| | 25% | 0.050 | 0.052 | 0.191 | 0.203 | 0.208 | 0.108 | 0.134 | 0.073 | 0.086 |
| | 40% | 0.057 | 0.058 | 0.200 | 0.217 | 0.237 | 0.126 | 0.153 | 0.081 | 0.103 |
| SAITS +MG | 10% | **0.037** | 0.032 | 0.197 | 0.194 | 0.146 | 0.075 | 0.105 | 0.043 | 0.032 |
| | 25% | 0.044 | 0.037 | 0.209 | 0.209 | 0.178 | 0.091 | 0.115 | 0.037 | 0.040 |
| | 40% | 0.055 | 0.046 | 0.234 | 0.244 | 0.227 | 0.116 | 0.148 | 0.067 | 0.057 |
| TimesNet +MG | 10% | 0.050 | 0.046 | 0.194 | 0.231 | 0.161 | 0.085 | 0.095 | 0.053 | 0.045 |
| | 25% | 0.052 | 0.054 | 0.201 | 0.238 | 0.192 | 0.100 | 0.111 | 0.062 | 0.059 |
| | 40% | 0.057 | 0.064 | 0.209 | 0.246 | 0.217 | 0.114 | 0.126 | 0.071 | 0.070 |
| iTransformer +MG | 10% | 0.075 | 0.055 | 0.166 | 0.185 | 0.194 | 0.112 | 0.128 | 0.067 | 0.052 |
| | 25% | 0.071 | 0.055 | 0.186 | 0.208 | 0.214 | 0.120 | 0.142 | 0.076 | 0.064 |
| | 40% | 0.073 | 0.060 | 0.205 | 0.231 | 0.234 | 0.130 | 0.156 | 0.084 | 0.076 |
| CSDI +MG | 10% | 0.040 | **0.019** | **0.102** | **0.104** | **0.100** | **0.045** | **0.065** | **0.027** | **0.019** |
| | 25% | **0.041** | **0.022** | **0.113** | **0.097** | **0.114** | **0.054** | **0.073** | **0.033** | **0.021** |
| | 40% | **0.042** | **0.024** | **0.122** | **0.110** | **0.136** | **0.062** | **0.082** | **0.038** | **0.024** |

Table 11: RMSE imputation results on real-world datasets under varying missing rates. The notation "+MG" signifies that the method incorporates our Meta Guidance framework.

| | | HD | Weather | Electricity | Traffic | ETTh1 | ETTh2 | ETTm1 | ETTm2 | TCPC |
|---|---|---|---|---|---|---|---|---|---|---|
| M-RNN | 10% | 0.821 | 0.468 | 0.490 | 0.527 | 0.826 | 0.779 | 0.839 | 0.737 | 0.763 |
| | 25% | 0.936 | 0.514 | 0.504 | 0.542 | 0.883 | 0.801 | 0.884 | 0.774 | 0.777 |
| | 40% | 1.019 | 0.555 | 0.519 | 0.564 | 0.920 | 0.824 | 0.920 | 0.803 | 0.805 |
| GAIN | 10% | 0.424 | 0.716 | 0.396 | 0.361 | 0.829 | 0.846 | 0.789 | 0.823 | 0.884 |
| | 25% | 0.418 | 0.677 | 0.392 | 0.383 | 0.821 | 0.849 | 0.826 | 0.850 | 0.925 |
| | 40% | 0.483 | 0.683 | 0.396 | 0.403 | 0.969 | 0.917 | 0.855 | 0.873 | 1.024 |
| BRITS | 10% | 0.235 | 0.162 | 0.357 | 0.463 | 0.372 | 0.210 | 0.180 | 0.134 | 0.121 |
| | 25% | 0.283 | 0.179 | 0.375 | 0.478 | 0.425 | 0.240 | 0.229 | 0.170 | 0.168 |
| | 40% | 0.333 | 0.197 | 0.405 | 0.504 | 0.488 | 0.279 | 0.271 | 0.212 | 0.237 |
| TIDER | 10% | 0.393 | 0.753 | 0.731 | 0.783 | 0.710 | 0.702 | 0.777 | 0.765 | 0.609 |
| | 25% | 0.445 | 0.760 | 0.738 | 0.795 | 0.768 | 0.748 | 0.829 | 0.805 | 0.611 |
| | 40% | 0.536 | 0.767 | 0.721 | 0.815 | 0.860 | 0.821 | 0.875 | 0.843 | 0.656 |
| Imputeformer | 10% | 0.185 | 0.195 | 0.370 | 0.488 | 0.412 | 0.211 | 0.202 | 0.120 | 0.158 |
| | 25% | 0.193 | 0.213 | 0.388 | 0.502 | 0.446 | 0.238 | 0.239 | 0.136 | 0.171 |
| | 40% | 0.205 | 0.229 | 0.430 | 0.589 | 0.512 | 0.266 | 0.295 | 0.160 | 0.195 |
| ModernTCN | 10% | 0.119 | 0.167 | 0.241 | 0.336 | 0.226 | 0.132 | 0.144 | 0.082 | 0.070 |
| | 25% | 0.119 | 0.184 | 0.272 | 0.382 | 0.248 | 0.138 | 0.159 | 0.088 | 0.077 |
| | 40% | 0.126 | 0.180 | 0.290 | 0.420 | 0.277 | 0.146 | 0.176 | 0.097 | 0.090 |
| PSW-I | 10% | 0.235 | 0.324 | 0.291 | 0.499 | 0.293 | 0.192 | 0.183 | 0.122 | 0.092 |
| | 25% | 0.255 | 0.334 | 0.325 | 0.504 | 0.315 | 0.199 | 0.192 | 0.129 | 0.107 |
| | 40% | 0.270 | 0.344 | 0.373 | 0.551 | 0.346 | 0.216 | 0.205 | 0.140 | 0.119 |
| Transformer | 10% | 0.188 | 0.182 | 0.379 | 0.441 | 0.333 | 0.242 | 0.257 | 0.202 | 0.248 |
| | 25% | 0.234 | 0.196 | 0.394 | 0.453 | 0.386 | 0.273 | 0.308 | 0.223 | 0.328 |
| | 40% | 0.247 | 0.204 | 0.407 | 0.466 | 0.428 | 0.302 | 0.344 | 0.242 | 0.382 |
| SAITS | 10% | 0.166 | 0.166 | 0.448 | 0.493 | 0.417 | 0.299 | 0.314 | 0.178 | 0.109 |
| | 25% | 0.202 | 0.179 | 0.455 | 0.510 | 0.530 | 0.340 | 0.411 | 0.220 | 0.133 |
| | 40% | 0.214 | 0.202 | 0.482 | 0.548 | 0.633 | 0.433 | 0.568 | 0.335 | 0.166 |
| TimesNet | 10% | 0.085 | 0.159 | 0.293 | 0.474 | 0.262 | 0.129 | 0.172 | 0.090 | 0.106 |
| | 25% | 0.096 | 0.170 | 0.302 | 0.477 | 0.311 | 0.150 | 0.200 | 0.102 | 0.131 |
| | 40% | 0.105 | 0.180 | 0.312 | 0.489 | 0.346 | 0.171 | 0.225 | 0.113 | 0.154 |
| iTransformer | 10% | 0.193 | 0.193 | 0.262 | 0.390 | 0.325 | 0.199 | 0.236 | 0.143 | 0.136 |
| | 25% | 0.300 | 0.214 | 0.298 | 0.433 | 0.368 | 0.227 | 0.269 | 0.173 | 0.164 |
| | 40% | 0.426 | 0.234 | 0.329 | 0.474 | 0.411 | 0.253 | 0.303 | 0.200 | 0.181 |
| CSDI | 10% | 0.163 | 0.151 | 0.188 | 0.356 | 0.187 | 0.085 | 0.124 | 0.065 | 0.057 |
| | 25% | 0.285 | 0.159 | 0.195 | 0.333 | 0.206 | 0.096 | 0.136 | 0.072 | 0.068 |
| | 40% | 0.265 | 0.172 | 0.205 | 0.358 | 0.242 | 0.110 | 0.151 | 0.079 | 0.074 |
| Transformer +MG | 10% | 0.082 | 0.159 | 0.261 | 0.388 | 0.255 | 0.130 | 0.169 | 0.089 | 0.118 |
| | 25% | 0.095 | 0.168 | 0.276 | 0.397 | 0.308 | 0.155 | 0.207 | 0.107 | 0.145 |
| | 40% | 0.105 | 0.176 | 0.289 | 0.422 | 0.350 | 0.181 | 0.235 | 0.117 | 0.169 |
| SAITS +MG | 10% | **0.081** | 0.161 | 0.286 | 0.401 | 0.226 | 0.113 | 0.164 | 0.072 | 0.069 |
| | 25% | 0.097 | 0.177 | 0.301 | 0.421 | 0.279 | 0.138 | 0.186 | 0.084 | 0.093 |
| | 40% | 0.111 | 0.182 | 0.332 | 0.462 | 0.354 | 0.174 | 0.234 | 0.105 | 0.121 |
| TimesNet +MG | 10% | 0.085 | 0.158 | 0.280 | 0.463 | 0.244 | 0.122 | 0.150 | 0.083 | 0.088 |
| | 25% | **0.094** | 0.169 | 0.292 | 0.471 | 0.287 | 0.142 | 0.176 | 0.094 | 0.111 |
| | 40% | 0.103 | 0.182 | 0.303 | 0.482 | 0.321 | 0.162 | 0.200 | 0.106 | 0.128 |
| iTransformer +MG | 10% | 0.109 | 0.176 | 0.241 | 0.375 | 0.292 | 0.157 | 0.205 | 0.099 | 0.103 |
| | 25% | 0.110 | 0.185 | 0.267 | 0.411 | 0.318 | 0.168 | 0.226 | 0.111 | 0.119 |
| | 40% | 0.118 | 0.192 | 0.293 | 0.448 | 0.344 | 0.183 | 0.244 | 0.122 | 0.138 |
| CSDI +MG | 10% | 0.107 | **0.127** | **0.163** | **0.263** | **0.177** | **0.078** | **0.110** | **0.056** | **0.054** |
| | 25% | **0.107** | **0.139** | **0.180** | **0.264** | **0.194** | **0.093** | **0.126** | **0.064** | **0.062** |
| | 40% | **0.103** | **0.152** | **0.195** | **0.301** | **0.228** | **0.103** | **0.142** | **0.074** | **0.068** |

## H Visualization of NSG and PG construction

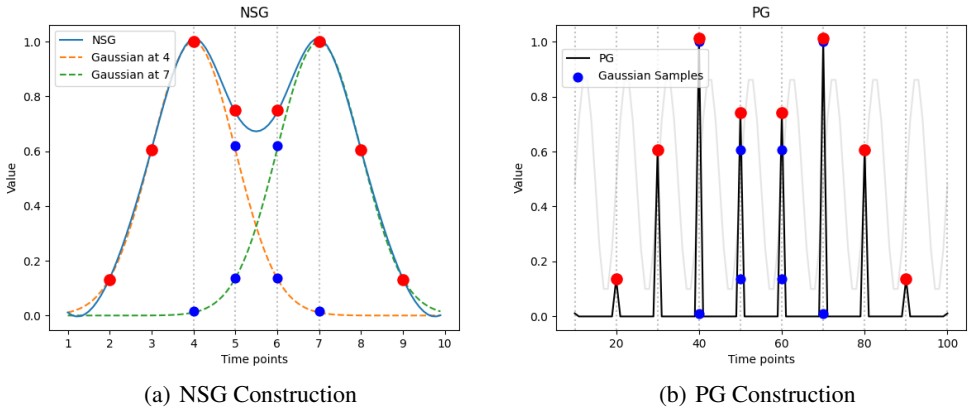

(a) NSG Construction                  (b) PG Construction

Figure 9: Visualizations of NSG and PG construction.

Figure 9(a) presents an illustrative example of NSG construction. In this example, the 4th and 7th points in the time series are missing. Given a hyperparameter setting of $r = 3$, each missing points will influence its three nearest neighbors, weighted by a Gaussian function $\psi(i)$. Specifically, if the 4th position is missing, the 3rd position receives a weight of $\psi(1)$, the 2nd position $\psi(2)$, and the 1st position $\psi(3)$, following the same pattern for other missing points. After applying this weighting scheme to all missing points, the resulting sequence constitutes the NSG.

Figure 9(b) presents an illustrative example of PG construction. The construction of PG follows a similar approach to NSG. In this example, the original time series has a period of $T = 10$, with missing values at the 40th and 70th positions. In PG, the influence of $\psi(i)$ propagates with a step size of $T$. Specifically, if the 40th position is missing, the 30th position receives a weight of $\psi(1)$, the 20th position $\psi(2)$, and the 10th position $\psi(3)$, following the same pattern for other missing points. After applying this weighting scheme to all missing points, the resulting sequence constitutes the PG.

## I Limitation

Due to its construction, Meta Guidance can only be integrated into deep learning methods and cannot be used with statistical and mathematical imputation methods. However, considering that Meta Guidance can significantly improve the performance of deep learning methods, even achieving state-of-the-art results, this limitation is acceptable.

## J Broader Impacts

Our work provides a more robust and accurate method for imputing missing values by integrating a novel framework Meta Guidance with existing deep learning architectures, crucial for data analysis in various domains. Furthermore, by addressing the challenges of non-stationarity and periodicity in time series data, this research can be instructive for future research in machine learning and data science, fostering the development of more sophisticated and adaptable imputation methods. Therefore, our paper mainly focuses on scientific research and has no obvious negative social impact.

