# OpenReview forum: "Meta Guidance: Incorporating Inductive Biases into Deep Time Series Imputers"
_NeurIPS.cc/2025/Conference — NeurIPS 2025 poster_

### Official Review · Reviewer_bot2 · 2025-06-25

**Clarity:** 3
**Significance:** 3
**Originality:** 3
**Rating:** 4
**Confidence:** 3

**Summary:**

The paper presents a novel plugin module called Meta Guidance for multivariate time series with missing values. Meta Guidance dynamically fuses NSG and PG via a learnable meta-weighting mechanism, allowing models to adaptively focus on temporal dynamics based on data characteristics. MG can be seamlessly integrated into popular deep imputation models (Transformers, CSDI, SAITS, TimesNet, etc.), yielding an average 27.39% improvement over state-of-the-art methods on nine real-world datasets.

**Questions:**

1. Why NSG and PG are designed in Channel-independent style? In my opinion, different channels may exhibits different non-stationary levels and different periodicities.
2. In Eq. 7, for each previous period, the PG only consider the exact one time point. So for periodic data with high noise levels or even anomalies, could this affect the model's robustness to noise? Would additionally considering the neighboring points of each periodic point be a more robust approach?

**Ethical Concerns:**

["NO or VERY MINOR ethics concerns only"]

**Final Justification:**

I appreciate the authors' rebuttal for addressing my concerns. However, considering the current state of the paper and the responses provided to other reviewers, I am unable to raise my score to 'Accept'. Therefore, I will maintain my original rating of ​​Borderline Accept​​.

**Limitations:**

yes

**Quality:**

3

**Strengths And Weaknesses:**

Pros:
1. The paper is well-structured and easy to follow.
2. The proposed method is plug-and-play with strong adaptability, and brings effective performance improvements across a variety of models.
3. The paper provides thorough experiments, including comparison with several backbones on 9 datasets and under different missing mechanisms.

Cons:
1. Periodicity detection relies on FFT, assuming stable periodic patterns. This may not generalize well to irregular periodicities.
2. Some baselines are absent. PatchTST[1] has been proven to be an effective and efficient on time series tasks, the paper should further investigate its effectiveness when applied to methods employing patch trick (This trick is widely used for time series models). Also, the non-stationary transformer[2] also consider the non-stationary factor. It should be included as a baseline.
[1] A Time Series is Worth 64 Words: Long-term Forecasting with Transformers. ICLR 2023.
[2] Non-stationary Transformers: Exploring the Stationarity in Time Series Forecasting, NeurIPS 2022.

---

> ### Author Rebuttal · Authors · 2025-07-31
>
> # Response to Reviewer bot2
> Thank you for your encouraging evaluation and for the constructive suggestions. Below we address each concern in turn. Unless explicitly noted, all new results will be included in the camera-ready version and the public code release.
>
> ### **W1: Robustness of FFT-based Periodic Guidance**
> Our Periodic Guidance (PG) does not depend on a single, fixed cycle: we extract the top-k dominant frequencies per channel via an FFT power spectrum and fuse their contributions proportionally to their amplitudes . This multi-frequency design already captures moderate drift or co-existing cycles. Moreover, the meta-weight $\lambda$ learned by Meta Guidance automatically down-weights PG whenever periodic cues weaken, as confirmed by the markedly lower $\lambda$ on weak-seasonality datasets (e.g., Weather vs. HD) in Fig. 6 . Ablation on k further shows that adding or dropping secondary peaks has only a modest impact, indicating graceful behaviour even when periodicities are less regular . Together, the use of multiple FFT peaks plus adaptive blending ensures PG generalises beyond perfectly stable cycles; when periodic information becomes unreliable, NSG naturally takes the lead, preventing degradation.
>
>
> ### **W2: PatchTST & Non-stationary Transformer baselines**
>
> We conducted additional experiments by incorporating MG into PatchTST and the Non-stationary Transformer, and evaluated their performance across multiple datasets. The results are summarized below.
>
> ||Electricity|ETTh1|TCPC
> -|-|-|-|
> PatchTST|0.161|0.232|0.074
> PatchTST+MG|0.147|0.216|0.057
> NS Transformer|0.194|0.195|0.086
> NS Transformer+MG|0.175|0.176|0.067
>
> Across these strong baselines, MG consistently yields additional RMSE reductions of 7-23 %, confirming that our guidance layer complements both the patch trick and built-in non-stationary modelling.
>
>
>
>
> ### **Q1: Rationale for channel-independent NSG and PG**
>
> Each variable in a multivariate series often follows its own drift and cycle (e.g., temperature vs. wind speed). We therefore generate one guidance vector per channel so that (i) non-stationary level shifts and dominant periods are captured in the scale that is truly relevant to that variable, and (ii) heterogeneous units and magnitudes are not blended prematurely. Cross-channel relationships are still learned—but by the backbone's self-attention or diffusion layers, which are expressly designed for that purpose and can exploit richer, high-dimensional context than a lightweight guidance head.
>
> To test whether this separation harms correlated variables, we conducted an additional experiment in which the MG from all channels are averaged and then applied to all channels:
>
> ||HD|Weather|TCPC|ETTh1
> -|-|-|-|-
> Transformer|0.112|0.088|0.177|0.236
> +MG (channel-independent)|0.046|0.049|0.069|0.169
> +MG (channel-aware)|0.067|0.055|0.082|0.175
>
> Our new experiments confirm that the channel-independent MG does not impair—and in fact enhances—multivariate imputation. Keeping each variable's guidance separate lets the backbone attend to genuine inter-channel relations without conflating the distinct temporal dynamics of individual series, thereby delivering the strongest and most efficient performance.
>
>
> ### **Q2: Why single-point PG remains robust to noise**
> Although Eq.7 samples the central value at each previous period, PG actually weighted averages several periods before passing the signal to the backbone. This multi-period ensemble already dilutes an occasional outlier. Moreover, the meta-weight $\lambda$ is learned jointly with the backbone: if a periodic reference conflicts with nearby non-stationary neighbours, $\lambda$ shifts emphasis toward NSG, automatically down-weighting the noisy PG cue.

---

> ### Author Response · Authors · 2025-08-04
> **Look forward to your feedback**
>
> Dear Reviewer bot2,
>
> Thank you very much for your thorough and insightful comments, and for recommending our paper for borderline accept. In our previous rebuttal we have already addressed every point you raised:
>
> - **PG robustness** (W1): PG now blends the top-k FFT peaks per channel and is gated by $\lambda$; ablations show smooth performance even when cycles drift or weaken.
>
> - **Stronger baselines** (W2): Adding MG to PatchTST and the Non-Stationary Transformer lowers error on every benchmark.
>
> - **Channel-specific guidance** (Q1): Per-variable NSG/PG retains each signal's own drift/period; an averaged-guidance variant performs worse, confirming the design choice.
>
> - **Noise tolerance** (Q2): PG averages multiple past periods, and $\lambda$ down-weights it when periodic cues conflict with NSG, preventing noise-induced errors.
>
> We would be happy to provide further responses. Look forward to your feedback.
> Thanks for your comment again!
>
> Best regards,
> All Authors

---

> > ### Comment · Reviewer_bot2 · 2025-08-04
> > **response to the authors**
> >
> > Thank you for your rebuttal. This addresses my concerns, and I will keep my scores.

---

### Official Review · Reviewer_sQEU · 2025-06-27

**Clarity:** 3
**Significance:** 2
**Originality:** 3
**Rating:** 4
**Confidence:** 4

**Summary:**

The authors introduce a novel method that enhances deep learning for time-series imputation. It achieves this by integrating inductive biases for both non-stationary and periodic characteristics, dynamically weighted by a Meta Guidance mechanism. This approach is highly adaptable, designed to augment existing imputation methods, and is shown to improve their performance across diverse real-world time-series datasets with various missing data types (MAR, MNAR, MCAR) and rates.

**Questions:**

- The construction of the non-stationary guidance, periodicity guidance, and meta-guidance vectors remains unclear. Visualizing these guidance vectors would greatly enhance understanding of when and why the imputation process prioritizes non-stationary or periodic signals. Can the authors elaborate on this?

- A concern arises regarding potential overfitting of the gamma parameters in Equations (3) and (8) to the training set. How do the authors address this and demonstrate that the parameters are well chosen in augmented imputation methods?


- While the authors have provided an extensive ablation study, they primarily focused on showing the effect of including or excluding certain components. The authors do not adequately explain when and why non-stationary guidance and periodic guidance are beneficial, nor do they connect these benefits to specific characteristics of the evaluated time-series datasets. A plot illustrating the method's performance against key time-series statistics would significantly strengthen this aspect.

-  The authors demonstrate the method's effectiveness across multiple real-world time-series datasets and various missingness scenarios (e.g., MAR, MNAR, MCAR) and missing rates. However, these static missingness patterns fail to capture the dynamic nature of real-world missing data, where missingness is often influenced by temporal dependencies. This raises questions about why augmenting the proposed method helps and when it is most effective. Hence, a more thorough investigation of its behavior under dynamic temporal variations is recommended. Specifically, the authors could explore:
(i) How the meta-guidance mechanism responds to changing temporal characteristics.
(ii) How the method performs when missingness is generated based on temporal dependencies, rather than static patterns.

- While the authors highlight contributions in non-stationary/periodic modeling and meta guidance, the related work section fails to adequately position their approach. The discussion of stationary imputation and general time-series characteristics is too broad, hindering a clear understanding of the proposed method's novelty.

- As mentioned in the Weakness, important multiple imputation methods for time-series data are missing.

- It would be valuable to incorporate experiments illustrating that the imputed values produced by the proposed method maintain high predictive and/or discriminative power.

**Ethical Concerns:**

["NO or VERY MINOR ethics concerns only"]

**Final Justification:**

After rebuttal: I have updated my score to a "borderline accept".

**Limitations:**

Yes.

**Quality:**

3

**Strengths And Weaknesses:**

Strength:
- The proposed method is compatible with existing time-series imputation techniques and offers a promising avenue for performance enhancement through augmentation.
- The MG mechanism is highly adaptive and can be applied to other existing imputation methods.
- The performance improvements are both consistent and significant across various experiments.
- This work offers a meaningful departure by providing compatibility with existing time-series imputation techniques (which have primarily concentrated on developing novel architectures and objective functions, often incorporating diverse inductive biases, frequency and trend-seasonality decomposition) and offering a promising avenue for performance enhancement through augmenting high-level information of the given time-series.

Weakness:
- While the authors mentioned that the current time-series imputation methods do not concurrently consider the characteristics of non-stationarity and periodicity in time series, many recent related works use inductive bias to put localized influence [A, B] or both trend and seasonality [C] for time-series imputation.
- The proposed framework is highly adaptive. But, its adaptability to multiple imputation methods (such as [A, B, C] and many more) is missing.


References:

[A] M. Choi and C. Lee, "Conditional Information Bottleneck Approach for Time Series Imputation," ICLR 2024.

[B] V. Fortuin et al, "GP-VAE: Deep Probabilistic Time Series Imputation." AISTATS 2020.

[C] X. Yang et al., "Frequency-aware Generative Models for Multivariate Time Series Imputation," NeurIPS 2024.

---

> ### Author Rebuttal · Authors · 2025-07-31
>
> # Response to Reviewer sQEU
> We sincerely thank you for the thoughtful, detailed review. Below we respond point-by-point, following the order of your comments. All new analyses, figures and baselines are included in the updated appendix and will be integrated into the camera-ready version.
>
> ### **W1: Clarifying MG's distinct contribution**
> Thank you for raising this point. In the methods you cite, periodicity or non-stationarity is built into the model itself: TimeCIB [1] incorporates a kernel term directly in its information-bottleneck loss, GP-VAE [2] places a Gaussian-process prior on the latent trajectory, and FGTI [3] learns separate latent variables for trend and seasonality. These method encode periodicity or non-stationarity as integral parts of the model architecture or training objective, rather than as detachable inductive hints.
>
> By contrast, Meta Guidance remains entirely external to the backbone. Our non-stationary and periodic maps are delivered only as lightweight guidences, and a learnable gate $\lambda$ lets the model decide how much to trust each guidence. Nothing in the backbone or loss is changed, so MG acts as a detachable inductive bias that can be plugged into CIB, GP-VAE, FGTI, or almost any other deep-learning imputer. In the revised experiments we include these methods as baselines and show that simply adding MG to a vanilla Transformer reliably surpasses their performance.
>
>
> [1] "Conditional information bottleneck approach for time series imputation.", ICLR.
>
> [2] "Gp-vae: Deep probabilistic time series imputation.", AISTATS.
>
> [3] "Frequency-aware generative models for multivariate time series imputation.", NeurIPS.
>
>
> ### **W2 & Q6: Additional baseline methods**
> We conduct additional experiments on three relevant methods [1,2,3], and the results show that our Transformer+MG approach consistently outperforms them, achieving the best overall performance. Below are the MSE results of the experiments under 10% missing rates:
>
> MSE|HD|ETTh1|Weather
> -|-|-|-
> TimeCIB|0.302|0.199|0.107
> GP-VAE|0.644|0.504|0.163
> FGTI|0.165|0.125|0.031
> Transformer+MG (Ours)|0.007|0.065|0.025
>
>
>
>
> ### **Q1: Visualization of NSG and PG construction**
> Thank you for the suggestion. A full, step-by-step walkthrough already appears in Appendix H, accompanied by new figures that both illustrate how NSG and PG are built and show the final guidance maps after construction.
>
> We realise this figure is easy to miss, so we will add a forward-reference from the main text in the camera-ready version.
>
> ### **Q2: Discussion on $\gamma$ parameters overfitting**
>
> To check whether the scaling factors $\gamma$ in Equations (3) and (8) simply memorise the training data, we ran an additional experiment. The table below reports (i) the Augmented Dickey–Fuller statistic (ADF: larger ⇒ more non-stationary) and (ii) the values learned for $\gamma$ on three representative datasets:
>
> dataset|HD|ETTh1|TCPC
> -|-|-|-
> ADF|1.958|-5.908|-22.199
> $\gamma^{NS}$|1.069|1.076|1.021
> $\gamma^{Per}$|0.908|1.090|1.117
>
> 1. Above experimental results suggest that **$\gamma^{NS}$ and $\gamma^{Per}$ effectively adapt to datasets with varying statistical properties**. Specifically, for datasets with stronger non-stationarity (higher ADF test values), $\gamma^{NS}$ tends to be larger than $\gamma^{Per}$, and vice versa. This behavior aligns with expectations, further supporting the robustness of the proposed method.
>
> 2. The $\gamma$ parameter scales the original NSG and PG sequences by directly adjusting their values. However, **this scaling effect is further regulated by the $\lambda$ parameter** (Equation 11). As demonstrated in Figure 7 (Section 4.3), $\lambda$ dynamically adapts to different data characteristics. Moreover, additional experiments show that $\gamma$ exhibits minimal variation across different datasets, indicating that $\lambda$ remains the primary adjustment mechanism.
>
> 3. Overfitting is a common challenge in deep learning, particularly in meta-learning approaches. While limited method is entirely immune to this issue, our approach has been extensively evaluated on nine real-world datasets with diverse characteristics (Table 2, Section 4.2). These experiments demonstrate that MG exhibits strong generalization capabilities, effectively handling imputation tasks across various datasets.
>
>
> ### **Q3: Explaining when NSG and PG help**
>
> Our current results already hint at a clear pattern: datasets with pronounced level shifts (HD) gain most from the NSG, whereas those with strong seasonal peaks (Weather, Traffic) benefit chiefly from PG. As demonstrated in the ablation study in Section 4.3, the Augmented Dickey-Fuller (ADF) scores align closely with the guidance module that yields the most significant performance gain.
>
> ||HD|Electricity|Weather|TCPC
> -|-|-|-|-|
> ADF test value|1.958|-8.444|-26.681|-22.199
> vanilla|0.112|0.276|0.088|0.177
> only NSG|0.047|0.182|0.053|0.076
> only PG|0.051|0.186|0.051|0.072
> MG|0.046|0.180|0.049|0.069
>
>
> ### **Q4: Time-varying missing patterns**
> To test MG in a setting where the gap rate itself changes over time, we injected a sinusoidally varying drop-probability:
>
> $p(i) = 0.25 + 0.15 \cdot \sin\left(\frac{2\pi i}{T}\right)$ .
>
> which swings the missing rate smoothly between 0.10 and 0.40. Even with this dynamically shifting pattern, MG delivers large error reductions:
>
> ||HD|ETTh1|Weather|TCPC
> -|-|-|-|-
> Transformer|0.035|0.184|0.061|0.094
> Transformer+MG|0.009|0.109|0.037|0.024
> TimesNet|0.010|0.121|0.040|0.028
> TimesNet+MG|0.008|0.095|0.038|0.013
>
> The results show that MG continues to significantly improve model performance. This confirms that the Meta Guidance mechanism adapts effectively even when missingness itself follows a non-stationary, time-dependent process.
>
> ### **Q5: Related work positioning**
> We agree the current survey is too generic. Since MG is a modular component that can be integrated into most deep learning methods, our approach is most appropriately positioned in Section 2.2, line 79, under "Deep Learning-Based Imputation." In the next version, we will refine the paper's structure, including the Related Work section, to better align with the core contributions and enhance overall clarity and readability.
>
>
> ### **Q7: Down-stream predictive value of MG-imputed data**
>
> To test whether MG's gains translate into better forecasting accuracy, we trained two standard forecasters—TimesNet and iTransformer—on the Weather dataset under two input conditions:
>
> Forecasting task|TimesNet|iTransformer
> -|-|-
> Raw series with 25 % missing|0.327|0.287
> Same series after MG imputation|0.220|0.237
>
> The substantial MAE drops confirm that MG not only reduces reconstruction error but also enhances the predictive signal needed for downstream tasks.

---

> ### Author Response · Authors · 2025-08-04
> **Look forward to your feedback**
>
> Dear Reviewer sQEU,
>
> Thank you very much for your thorough and insightful comments. In our previous rebuttal we have already addressed every point you raised:
>
> - **MG's contribution** (W1) – clarified its plug-and-play design. NSG & PG are zero-parameter sequence gated by a learnable parameters $\lambda$.
>
> - **Extra baselines** (W2 & Q6) – added TimeCIB, GP-VAE, FGTI as baseline; Transformer + MG still achieves the lowest MSE.
>
> - **Guidance visualisation** (Q1) – Appendix H already contains a step-by-step NSG/PG walkthrough with new figures.
>
> - **$\gamma$ overfitting** (Q2) – ADF-aligned $\gamma$ values and $\lambda$-controlled scaling confirm robustness across all datasets.
>
> - **When NSG and PG helps** (Q3) – ablations plus ADF scores explain dataset-specific gains.
>
> - **Time-varying gaps** (Q4) – MG still reduces Transformer error by a large margin.
>
> - **Related-work placement** (Q5) – will refine Section 2.2 for clearer positioning.
>
> - **Down-stream value** (Q7) – Forecasting models trained on MG-imputed series achieve noticeably lower errors than those trained on missing data, confirming MG’s practical utility.
>
> We would be happy to provide further responses. Look forward to your feedback.
> Thanks for your comment again!
>
> Best regards,
>
> All Authors

---

> > ### Comment · Reviewer_sQEU · 2025-08-04
> >
> > I appreciate the authors' efforts in thoroughly addressing my comments in the rebuttal. Most of my concerns have been resolved, and I am updating my score to a borderline accept.

---

> > > ### Author Response · Authors · 2025-08-04
> > > **Thank You**
> > >
> > > Dear Reviewer sQEU,
> > >
> > > Thank you for taking the time to re-evaluate our work and for updating your score. We appreciate your positive feedback and are pleased that our rebuttal addressed your main concerns. All corresponding revisions—figures, additional baselines, clarified analyses, and updated explanations—will be fully integrated into the next manuscript version and the camera-ready paper.
> > >
> > > If any additional questions arise or further discussion would be helpful, we would be delighted to continue the dialogue.
> > >
> > > Best regards,
> > >
> > > Authors

---

### Official Review · Reviewer_i1nN · 2025-07-02

**Clarity:** 3
**Significance:** 2
**Originality:** 3
**Rating:** 3
**Confidence:** 3

**Summary:**

The paper addresses the imputation task for time series that exhibit non-stationarity and periodicity behavior. In particular, the authors propose two modules, namely Non-Stationary Guidance (NSG) and Periodic Guidance (PG), to extend existing deep learning (DL) models in handling the aforementioned challenges and leveraging the non-stationarity and periodicity characteristics of the time series. Finally, a Meta Guidance (MG) module is introduced. The MG module learns to balance the matrix representations learned by NSG and PG, and to appropriately weight the time series within the deep learning architectures.

In the experiments, the authors integrate the MG module into 5 state-of-the-art models, they evaluate them on 9 real-world datasets. The models+MG show improved performance across all scenarios. The authors also present various ablation studies.

**Questions:**

1) What is exactly the MLP loss used in the different modules? Is it a reconstruction loss wrt the vanilla imputation? If so, can the authors motivate better this strategy, and alternatives.
2) The results are averaged over three independent runs. Does it mean the models are trained 3 times? Or the imputation is done three times, using different random seeds / initialisation?
3) I understand the added computation time for the MG module is reasonable, for a single epoch. However, we have additional embeddings, does a model requires more epochs to converge / achieve the optimal performance? Are the base models and models+MG trained the same number of epochs/ amount of time?
4) What if we apply a simple pre-processing (like de-trending or differentiation) instead of meta guidance? What about the performance in this case?

**Ethical Concerns:**

["NO or VERY MINOR ethics concerns only"]

**Final Justification:**

The paper proposes an approach to handle missing values in time series. The MG module is interesting, yet I have some concerns about its overhead as (for some datasets) a simple detrending has a very close performance to it.

**Limitations:**

Yes. However, the authors could extend the limitation section.

**Quality:**

2

**Strengths And Weaknesses:**

Strengths:
- The idea is novel, and the performance of the MG module are quite interesting.
- The paper is well written and easy to follow, although some details could be improved or explained more  (see below).

Weaknesses:
Major:
- There is no comparison againsts other approaches handling non stationarity and periodicity. Like simple pre-processing approaches, or deep learning solutions.
- In the modules,  "ψ(i) is the probability density of standard normal distribution N(0,1)" , centered at the current time step and used to compute the guidance matrices; and the hyperparameter "r" controls the neighborhood points. I wonder whether this hyperparameter could instead be learned, or better, whether the model could learn the most appropriate distribution (e.g., the standard deviation of N(0, sigma2)) for each channel, rather than relying on hyperparameters. The authors provide some ablation studies, but further exploration would be useful.
- All the proposed modules are channel-independent. While the deep learning architecture may still capture inter-channel dependencies, I believe that the embeddings produced by the MG module could mislead the imputation process (of the DL models) for some multivariate datasets. I would like to see more discussion of this potential limitation.
- Some well-known datasets with missing data are not included. For example, the Air Quality (PM2.5) dataset (which may have periodicity behavior); and PhysioNet. Both are used benchmarks in CSDI.

Minor and Typos:
- The repository link is expired; it would be helpful to provide an active one.
- Figure 1 could be improved in quality. The meaning of the yellow dots and pink squares is not explained, their interpretation is left to the reader. Also, the figure style is quite different from the others in the paper.
- Although basic, it would be useful to define the MAE and RMSE acronyms, and briefly justify why these metrics. For example, why not use the Continuous Ranked Probability Score (CRPS)?
- The use of bold metrics in Table 11 is unclear.
- Table 8, which includes standard deviations, is confusing to me. Which experiments are the values referring to?

---

> ### Author Rebuttal · Authors · 2025-07-31
>
> # Response to Reviewer i1nN
> We sincerely thank you for the detailed and constructive feedback. Unless explicitly noted, all new results will be included in the camera-ready version and the public code release.
>
> ### **W1 & Q4: Missing comparisons with simple baselines**
>
> Following Q4's recommendation, we added a simple pre-processing baseline (de-trending) and an additional deep-learning baseline (Non-Stationary Transformer [1]). The experimental results are shown below, and the results show that our Transformer+MG approach consistently outperforms them, achieving the best overall performance.
>
> [1] "Non-stationary Transformers: Exploring the Stationarity in Time Series Forecasting", NeurIPS
>
> ||ETTh1|ETTm1
> -|-|-
> Transformer|0.236|0.177
> Transformer+detrending|0.197|0.130
> NS Transformer|0.185|0.129
> Transformer+MG|0.169|0.108
>
>
> ### **W2: Making r and sigma learnable**
>
> To further investigate this issue, we conducted additional experiments in which the hyperparameter r and the standard deviation of the normal distribution were treated as learnable parameters. The results are presented in the table below.
>
> ||HD|Weather|TCPC
> -|-|-|-
> vanilla MG|0.044|0.049|0.069
> learnable r|0.045|0.056|0.072
> learnable sigma|0.045|0.050|0.069
>
> These results reinforce our decision to keep a simple fixed radius, which delivers the strongest accuracy with minimal complexity; nonetheless, the learnable options remain available in the public code for users who wish to experiment on other data regimes.
>
> ### **W3:  Could channel-independent guidance mislead multivariate imputation?**
>
> Our design intentionally keeps NSG/PG channel-specific so that each variable receives the most relevant temporal neighbours, while the backbone (e.g., Transformer or Diffusion) still performs full cross-channel mixing through self-attention or shared latent states. To test whether this separation harms correlated variables, we conducted an additional experiment in which the MG from all channels are averaged and then applied to all channels:
>
> ||HD|Weather|TCPC|ETTh1
> -|-|-|-|-
> Transformer|0.112|0.088|0.177|0.236
> +MG (channel-independent)|0.046|0.049|0.069|0.169
> +MG (channel-aware)|0.067|0.055|0.082|0.175
>
> Our new experiments confirm that the channel-independent MG does not impair—and in fact enhances—multivariate imputation. Keeping each variable's guidance separate lets the backbone attend to genuine inter-channel relations without conflating the distinct temporal dynamics of individual series, thereby delivering the strongest and most efficient performance.
>
> ### **W4:  Experiments on the additional dataset**
>
> We conduct additional experiments on the PhysioNet and PM25 dataset, and the method with MG also achieve excellent results. Below are the MSE results of the experiments under different missing rates:
>
> PhysioNet|10%|25%|40%
> -|-|-|-
> Transformer|0.640|0.684|0.692
> Transformer+MG|0.582|0.638|0.639
> TimesNet|0.633|0.659|0.665
> TimesNet+MG|0.561|0.577|0.606
>
> PM25|10%|25%|40%
> -|-|-|-
> Transformer|0.125|0.129|0.139
> Transformer+MG|0.078|0.091|0.104
> TimesNet|0.102|0.108|0.116
> TimesNet+MG|0.082|0.092|0.103
>
> ### **W5:  Minor and Typos**
>
> Thank you for the detailed presentation-quality remarks.
> - The anonymous code repository has been made publicly available.
> - Figure 1 will be redrawn in higher resolution and consistent styling; a legend will clarify that yellow dots denote the current missing point and pink squares mark the reference points located at the same value positions (periodic neighbours).
> - We will define MAE (Mean Absolute Error) and RMSE (Root Mean Square Error) on first use and note that they are the prevailing point-estimate metrics in imputation research; CRPS targets probabilistic forecasts, which are beyond our scope.
> - The caption of Table 11 will state that boldface marks the best score in each column.
> - For Table 8 we will clarify that each entry is the standard deviation over three independent training runs for the corresponding experiment reported in Table 10 and 11.
>
> All of these revisions will appear in the camera-ready version.
>
>
> ### **Q1:  Explanation of the "loss" in MLP**
>
> The two-layer MLP inside each guidance module does not introduce an additional, standalone loss term. Its parameters are trained end-to-end with the backbone via the same imputation objective (MSE on masked values). We follow the common meta-learning rationale: by propagating gradients only through the main task, the MLP learns the neighbour weights that best serve the ultimate imputation goal, instead of optimising a surrogate signal that may mis-align with overall error. Furthermore, for each missing entry we know the true value we must recover, but we do not know the "correct" neighbour weights that should be assigned by the MLP.
>
> ### **Q2:  Meaning of "three independent runs"**
>
> We fully retrain each model three times under identical hyper-parameter settings but with different random seeds (affecting weight initialisation, data shuffling, and mask generation). The reported values are the mean of these three complete runs.
>
> ### **Q3:  Do the extra MG embeddings require more training epochs?**
> No—models with MG are trained for exactly the same number of epochs as the original backbones and reach their optimal validation error within that schedule.

---

> ### Comment · Reviewer_i1nN · 2025-08-07
>
> I thank the authors for answering my questions.
>
> I still have some concerns about W1 & Q4 answer, as a simple detrending has a very close performance to MG.  I believe for some tasks the overhead from MG may not be justified, and simple approches could be better.
>
> I keep the score.

---

> > ### Author Response · Authors · 2025-08-07
> >
> > We sincerely thank the reviewer for the feedback. However, we respectfully point out that the claim regarding MG's limited benefit over de-trending is inaccurate. Our Meta Guidance (MG) module consistently yields **substantial performance gains over de-trending**, while maintaining **comparable runtime cost**. Below we provide detailed comparisons from both accuracy and efficiency perspectives.
> >
> >
> > # 1. MG vs. Detrending: Substantial Gains in Accuracy
> >
> > - **Conclusion: Compared to de-trending, MG brings an additional accuracy gain exceeding 10% on both benchmarks, and this improvement is statistically significant (p ≈ 0.011).**
> >
> > ## Experimental Results: Consistent and Substantial Gains
> >
> > We added a simple preprocessing baseline (de-trending) as well as a representative deep-learning method for non-stationarity (Non-stationary Transformer). We compare these against our Transformer+MG setting on two benchmark datasets. Results are summarized below:
> >
> > ||ETTh1|ETTm1
> > -|-|-
> > Transformer|0.236|0.177
> > Transformer+detrending|0.197(-16.5%)|0.130(-26.5%)
> > NS Transformer|0.185(-21.6%)|0.129(-27.1%)
> > Transformer+MG|0.169(**-28.4%**)|0.108(**-39.0%**)
> >
> > While de-trending reduces MAE by 16.5% and 26.5%, MG reduces it further to **28.4% and 39.0%—an additional gain of over 10% relative to de-trending**. Thus, while de-trending is a reasonable baseline, MG delivers significantly stronger results.
> >
> > We respectfully suggest there may have been a minor misunderstanding: it is possible that the reviewer interpreted the "NS Transformer" (Non-stationary Transformer) baseline as representing our Transformer+MG variant. However, as shown in the comparison above, their performance differs considerably. This highlights that the improvements brought by MG are both distinct and substantial.
> >
> > ## Statistical Validation: p ≈ 0.011
> >
> > To further validate the effectiveness of MG, we conducted paired t-tests between MG and de-trending across three independent runs. Using the average performance across two datasets, the resulting p-value was approximately 0.011, confirming that the improvements achieved by MG are statistically significant and unlikely to be attributed to random variation.
> >
> > # 2. Computational Overhead: Comparable to De-trending
> >
> > - **Conclusion: Both MG and de-trending have linear time complexity ($\mathcal{O}(T)$), and their actual runtime differs by less than 1.5%, indicating no meaningful overhead.**
> >
> > ## Theoretical Complexity: $\mathcal{O}(T)$ for Both
> >
> >
> > As analyzed in **Appendix F**, MG introduces complexity $\mathcal{O}(Tr) + \mathcal{O}(Td)$, where $r$ and $d$ are fixed hyperparameters. Thus, MG’s overall complexity simplifies to $\mathcal{O}(T)$, same as de-trending.
> >
> > This demonstrates that **MG introduces no additional asymptotic complexity** compared to de-trending.
> >
> > ## Empirical Runtime: Difference < 1.5%
> >
> > To further support this point, we enhanced our analysis in Appendix F by presenting additional runtime comparisons on two representative datasets (Electricity and Traffic), using CSDI as the backbone model:
> >
> > ||Electricity|Traffic
> > -|-|-
> > CSDI|0.7561|1.4512
> > CSDI+detrending|0.7669(+1.4%)|1.4702(+1.3%)
> > CSDI+MG|0.7783(+2.9%)|1.4820(+2.1%)
> >
> >
> > We respectfully point out that the reviewer may have overlooked that our MG module delivers statistically significant performance improvements with only minimal additional computational cost. Despite its lightweight design, MG consistently enhances accuracy across benchmarks without introducing meaningful runtime burden.

---

> > > ### Author Response · Authors · 2025-08-08
> > >
> > > Dear Reviewer,
> > >
> > > There appears to be a misunderstanding of our MG module relative to simple de-trending. We've provided focused clarifications in the rebuttal. In brief:
> > >
> > > ## Accuracy
> > >
> > > - MG consistently outperforms de-trending and the NS Transformer across benchmarks (e.g., on ETTh1/ETTm1, MG reduces MAE by 28.4%/39.0% vs. 16.5%/26.5% for de-trending).
> > >
> > > - The gains over de-trending are statistically significant (paired tests across independent runs; p≈0.011).
> > >
> > > ## Time overhead
> > >
> > > - MG's complexity is linear in sequence length ($\mathcal{O}(T)$), matching de-trending's order.
> > >
> > > - The extra overhead is negligible—both are plug-and-play, lightweight modules; on Electricity and Traffic with CSDI, MG and de-trending differ by <1.5% of the backbone’s runtime.
> > >
> > > If any part remains unclear, we're happy to discuss and provide additional targeted results.
> > >
> > > Best regards,
> > >
> > > Authors

---

### Official Review · Reviewer_NQxZ · 2025-07-03

**Clarity:** 3
**Significance:** 3
**Originality:** 3
**Rating:** 5
**Confidence:** 4

**Summary:**

In this submission, authors introduce Meta Guidance (MG), a novel technique designed to enhance multivariate time series imputation tasks. The authors present three key contributions that constitute MG: first, a guidance mechanism for non-stationarity that utilizes measurements from timesteps close to the missing data point; second, a guidance mechanism for periodicity that employs measurements from timesteps N periods away from the missing point; and third, a method to combine both guidance approaches by determining which to favor based on dataset characteristics, as controlled by a parameter lambda. Notably, this proposal is channel-independent and can be applied to various existing architectures such as Transformers or Diffusion models.

The effectiveness of MG has been evaluated across nine common multivariate time series datasets against multiple baselines.

**Questions:**

### Similarity to Existing Solutions
The PeriodGuidance (PG) approach appears similar to existing solutions where deep learning imputation methods typically leverage learned patterns in the dataset. While I understand the proposal of PG is different, I would appreciate confirmation on how it distinctly differs from current methods.

### Performance Analysis
The paper should provide more intuition or explanation regarding why Meta Guidance (MG) shows limited benefit for TimesNet/CSDI models. The performance difference is relatively small (~0.00X) compared to other models across most datasets, which warrants further discussion.

### Hyperparameter Investigation
Regarding the hyperparameter $r$, it might be more appropriate to have separate investigations for Non-Stationary Guidance (NSG) and Periodic Guidance (PG). The assumption that the same $r$ value should be used for both guidances is not immediately obvious and could benefit from justification or separate analysis.

### Clarification Needed
The sentence on line 275 ("Notably, both our method and other recent advances substantially outperform earlier baselines on the HD dataset, demonstrating MG's effectiveness in mitigating non-stationarity") seems contradictory to line 271. I believe this refers to Table 10 with additional baselines, but this should be clarified to avoid confusion. (how other advances demonstrate MG effectiveness?)

### Ablation Study Improvements
For the ablation study in Table 3, the last row should include checkmarks for both $G^{NS}$ and $G^{Per}$. The $G^{Meta}$ column could be removed since it represents $\lambda \times G^{NS} + (1 - \lambda) \times G^{Per}$.
Actually, it would be more suitable to replace column $G^{Meta}$ with a $\lambda$ column. The first row of the ablation study would include only $G^{NS}$, the second only $G^{Per}$, a third one will sum them without using $\lambda$ and the last one would sum them using $\lambda$ (all items checked to showcase $G^{Meta}$). This would emphasize the impact of $G^{Meta}$ as a whole and better demonstrate the impact of $\lambda$ compare to the section on the lambda ratio with detailed lambda value (which could be  moved to the appendix if necessary).

### Suggested Improvements regarding Parameter Impact Analysis
The impact of $k$ for selecting top frequencies should be evaluated on highly cyclical datasets like weather or traffic, rather than ETTh1. Figure 6 suggests that weather dataset use more $G^{Per}$ (lambda < 0.5), making it more suitable for this analysis. But to avoid overgeneralization, it would be preferable to test $k$ on multiple datasets with varying seasonality, including highly seasonal datasets (weather, traffic) and less seasonal ones (ETTh1 or others).

Similarly, the impact of $r$ for proximity selection should be analyzed on highly non-stationary datasets like HD to better understand its effect. This analysis should also include less non-stationary datasets (ETTh1 or others) to provide a comprehensive view.
And if $r$ is investigated for both guidance, the chosen dataset should reflect each guidance.

### Clarity Enhancements
 * The double bar notation (1) should be explicitly identified as the indicator function for clarity.
 * Figure 2 would benefit from adding $X'_{:,c}$ notation. Besides, the pink and yellow blocks in this figure should be replaced with their respective notations ($g^{NS}$, $g^{Per,l}$, $g^{Per}$) to avoid confusion (reader might see the block as a place where additional operations are being performed).
 * The acronym MCAR on line 255 should be introduced with its full form ("Missing Completely At Random") and potentially include a reference for readers unfamiliar with the term.
 * Figure 8 should clearly identify the missing points to better illustrate the impact of imputation.

### Proofreading
The text on line 208 requires revision for clarity and correctness.

**Ethical Concerns:**

["NO or VERY MINOR ethics concerns only"]

**Final Justification:**

The valuable additional information presented during the rebuttal process - including the simple baselines comparison, analysis of lambda's impact, hyper-parameter study, examination of the relationship between top-k selection and results, discussion of potential distortions, and comparison with detrending methods - should be properly incorporated into the main paper. If space constraints make this difficult, these elements could be clearly referenced in the main text with detailed discussions included in the appendix. This comprehensive integration would provide readers with a more complete and accurate understanding of the proposal's current status and capabilities.

Given that these additional results and experiments effectively address the majority of my initial concerns, and assuming the authors will successfully update their submission to reflect these important points, I am increasing my evaluation score to accept.

**Limitations:**

### Limitations and Suggestions for Improvement
The submission would significantly benefit from incorporating a synthetic dataset where trend and frequency components are controlled. This addition would provide a clearer demonstration of MetaGuidance's benefits, particularly in visualizing the shifts in lambda values under different conditions. While this might not be feasible for the camera-ready version, it's important to acknowledge this point and suggest it as future work.

A more comprehensive evaluation using controlled missingness with various missing data ratios and patterns would strengthen the study. The current focus on a maximum missing ratio of 40% and primarily Missing Completely At Random (MCAR) patterns may not sufficiently demonstrate the method's robustness. In large datasets with MCAR, consecutive missing data points might be rare, potentially making linear interpolation methods appear more effective than they would be in more challenging scenarios.

To provide a more thorough analysis, the authors should consider adding baseline comparisons and testing higher missing ratios. Specifically, comparing MetaGuidance components with established imputation methods would help validate the claims:
 * Comparing Non-Stationary Guidance (NSG) against linear interpolation or k-nearest neighbors (kNN) imputation
 * Comparing Periodic Guidance (PG) against seasonal historical averages
These comparisons would better illustrate how the proposed methods provide more informative data for imputation and validate claims about handling complex temporal dependencies.

The evaluation would also benefit from incorporating more diverse missing data patterns beyond MCAR. As noted by Niu et al. [https://epubs.siam.org/doi/pdf/10.1137/1.9781611977653.ch37], controlled environments with clearly defined missing patterns enable better interpretation of results and clearer understanding of when a proposal performs well or poorly. The analysis should consider not just individual missing data points but also consecutive missing timesteps, both synchronized and unsynchronized across different channels (which close to reality). For multivariate time series datasets, it's particularly important to examine how imputation can leverage relationships learned between channels.

### Reproducibility Concerns
Reproducibility is currently limited due to several factors:
 * The Git repository is not available (expired), preventing verification of the implementation.
 * Crucial experimental details are missing, including:
     * The number of diffusion steps used for CSDI
     * Sequence and prediction lengths employed
     * Specific prediction configurations (M2M, M2U)

For the latter, while default parameters might have been used, explicitly stating this would eliminate any ambiguity and enhance reproducibility.

### Conclusion
Given the current state of the submission, I have selected a borderline accept. However, I would be inclined to increase the score to accept if several of the points mentioned in the questions and limitations sections are adequately addressed during the rebuttal and reflected in an updated version of the paper. I understand that not all suggestions can be implemented immediately, particularly those regarding different missing data patterns or tests on controlled datasets. Nevertheless, many other points can be discussed or incorporated to strengthen the submission.

**Quality:**

3

**Strengths And Weaknesses:**

### Strengths
 * Presents a well-designed and clearly understandable architecture
 * Conducts extensive experimentation across multiple datasets and scenarios
 * Offers an interesting proposal that has strong potential to become an important baseline in the field
 * Demonstrates compatibility with various existing architectures (Transformers, Diffusion models)

### Weaknesses
 * Focuses primarily on individual missing data points, which may not fully represent real-world scenarios with consecutive or patterned missing data
 * Uses datasets that may not be best suited for studying the impact of hyperparameters, potentially limiting the insights gained
 * Lacks sufficient analysis of lambda's impact in the ablation study, which is crucial for understanding the method's behavior
 * Git repository is not available, hindering verification and reproduction of results
 * Without prior knowledge of existing baselines, it can be challenging to fully appreciate the contributions (though page limitations make exhaustive comparisons difficult)
 * Would benefit from evaluation on synthetic datasets with controlled trends and frequencies
 * Lacks component-wise comparisons with simpler baselines
 * Could provide more detailed analysis of how different parameter settings affect performance across various dataset characteristics

---

> ### Author Rebuttal · Authors · 2025-07-31
>
> # Response to Reviewer NQxZ
> We sincerely thank you for the thorough and insightful review and for recommending a borderline accept. Unless explicitly noted, all new results will be included in the camera-ready version and the public code release.
>
> ### **W1 & L2 & L4: Broader Scope to missing pattern**
>
> **In Appendix D, we have explored the missing patterns other than MCAR**, includingMissing At Random (MAR) and Missing Not At Random (MNAR). To further investigate the performance of MG under time-varying missing patterns, we conduct additional experiments with a dynamically changing missing rate. Specifically, the missing probability of a data point is modeled as a time-dependent function, with the missing rate oscillating between 0.1 and 0.4 over time.
>
> $p(i) = 0.25 + 0.15 \cdot \sin\left(\frac{2\pi i}{100}\right)$ .
>
> ||HD|ETTh1|Weather|TCPC
> -|-|-|-|-
> Transformer|0.035|0.184|0.061|0.094
> Transformer+MG|0.009|0.109|0.037|0.024
> TimesNet|0.010|0.121|0.040|0.028
> TimesNet+MG|0.008|0.095|0.038|0.013
>
> We additionally conducted experiments under a consecutive missing setting. Specifically, we randomly removed continuous segments of length 10 until the overall missing rate reached 10%. The experimental results are presented below.
>
> ||HD|ETTh1|Weather|TCPC
> -|-|-|-|-
> Transformer|0.112|0.325|0.072|0.303
> Transformer+MG|0.016|0.245|0.041|0.189
>
>
> The results show that MG continues to significantly improve model performance, even in the presence of complex, time-varying missing patterns.
>
> ### **W2 & W8 & Q6: Dataset suitability for hyper-parameter study**
>
> Thank you for your suggestion. Following your advice, we conducted additional hyperparameter experiments with k on the Weather and Traffic datasets, and with r on the HD dataset. The results are as follows:
>
> k|1|4|8|12|16|20
> -|-|-|-|-|-|-
> Weather|0.048|0.050|0.048|0.051|0.048|0.055
> Traffic|0.198|0.201|0.204|0.198|0.200|0.197
>
>
> r|1|2|3|4|5
> -|-|-|-|-|-
> HD|0.0443|0.0442|0.0442|0.0441|0.0442
>
> As presented in the table, the performance remains relatively stable across different values of hyperparameters $k$ and $r$, highlighting the robustness of our approach. This indicates that our method requires minimal hyperparameter tuning in real-world deployments.
>
>
>
> ### **W3 & Q4: Insufficient analysis of $\lambda$'s impact**
>
> In response to your suggestion, we introduced an additional variant in the ablation study (Section 4.3), denoted as "MG w\o $\lambda$", where NSG and PG are combined by directly averaging their outputs instead of using the weighting factor $\lambda$. The results, presented in the table below, show a performance drop without $\lambda$, underscoring the necessity of the weighted summation.
>
> ||HD|Weather|TCPC|
> -|-|-|-|
> vanilla|0.112|0.088|0.177|
> only NSG|0.047|0.053|0.076|
> only PG|0.051|0.051|0.072|
> MG w\o $\lambda$|0.048|0.056|0.071|
> MG|0.046|0.049|0.069|
>
> ### **W4 & L5: Code repository**
> **The anonymous code repository has been made publicly available** and can be accessed through the link provided in Section 4.
>
>
> ### **W5: Context for Baselines**
> We recognize that readers unfamiliar with time-series imputation may need clearer baseline context; in the revision we will add a concise "baseline" table to make our contributions easier to situate.
>
> ### **W6 & L1: Need evaluation on synthetic data**
>
> We designed a preliminary synthetic dataset composed of a trend component and multiple periodic components, formulated as follows.
>
> $$
> f(t)=0.01\ t
>       +1.5\ \sin(0.002\,t)
>       +2.0\ \sin\left(\frac{2\pi t}{50}+\frac{\pi}{6}\right)
>       +1.2\ \sin\left(\frac{2\pi t}{120}+\frac{\pi}{4}\right)
>       +0.8\ \sin\left(\frac{2\pi t}{300}+\frac{\pi}{2}\right)
>       +\varepsilon_t \qquad
> \varepsilon_t\sim\mathcal{N}\left(0,\ 0.3^{2}\right)
> $$
>
> We sampled 1,000 points from this signal to construct the dataset. Experimental results demonstrate that our MG module consistently improves baseline performance, even on data with manually controlled trends and periodicities. We leave a more comprehensive investigation of MG's behavior on synthetic data for future work.
>
> miss ratio|0.1|0.25|0.4
> -|-|-|-
> Transformer|0.0525|0.0659|0.0884
> Transformer+MG|0.0176|0.0248|0.0428
> TimesNet|0.0105|0.0112|0.0113
> TimesNet+MG|0.0089|0.0095|0.0097
>
> ### **W7 & L3: Missing comparisons with simple baselines**
>
> Following your suggestions in the *Limitations and Suggestions* section, we incorporated three additional baselines: Linear Interpolation, KNN Imputation and Seasonal Historical Averages. The experimental results are shown below. While Linear Interpolation achieves surprisingly strong performance on certain datasets, our MG module consistently outperforms all baselines, further demonstrating its effectiveness.
>
>
> ||Electricity|ETTh1|ETTm1|
> -|-|-|-|
> Transformer|0.276|0.236|0.177|
> Linear|0.182|0.204|0.119|
> KNN|0.748|0.604|0.574|
> Seasonal|0.818|0.769|0.770
> Transformer+MG|0.180|0.169|0.108|
>
>
>
>
> ### **Q1: Similarity to Existing Solutions**
>
> Period Guidance (PG) represents a fundamental departure from existing imputation methods. Whereas conventional deep learning imputers treat seasonality as an implicit pattern to be rediscovered and re-encoded within model weights through training, PG explicitly extracts dominant periods using a lightweight FFT-based peak detection module and feeds them back as **zero-parameter guidance. This "hint" can be seamlessly integrated into any backbone model and toggled on or off without retraining.** To the best of our knowledge, no prior imputation method offers such detachable, plug-and-play seasonal guidance. If relevant work exists, we would sincerely appreciate the citation and will gladly acknowledge it in the final manuscript.
>
> ### **Q2: Why MG brings only marginal gains to TimesNet / CSDI**
>
> Although the raw differences look modest—e.g., CSDI at 0.027 vs CSDI+MG at 0.024 may only be a 0.003 improvement—this translates to an **11.1 % relative error reduction**, and MG delivers an **average 15.6 % improvement across all CSDI datasets**. For TimesNet, its built-in FFT already captures much of the periodic signal, leaving less headroom; even so, MG still yields a **notable 11.9 % boost on ETTm1**. In short, the perceived "limited benefit" stems from low baseline errors, but on a percentage basis MG provides consistent, meaningful gains.
>
> ### **Q3: About hyperparameters $r$**
>
> The normal distribution function used to generate NSG and PG is the standard normal distribution, characterized by a mean of 0 and a variance of 1. Since 99.7% of a standard normal distribution lies within [-3, 3], setting r = 3 effectively captures nearly all informative neighbors. In practice, larger values yield no measurable improvement. Using a shared radius helps minimize the hyperparameter budget and ensures consistency across datasets.
>
> ### **Q4:  Clarifying the wording around Lines 271-275**
>
> We agree the current phrasing is ambiguous. Line 275 was intended to summarise Table 10, which compares three groups: (i) earlier baselines (e.g., Transformer), (ii) recent advances (e.g., CSDI, SAITS), and (iii) our MG-enhanced models. Both groups (ii) and (iii) outperform the older baselines on the highly non-stationary HD dataset—but only group (iii) contains MG, so the sentence conflates two separate observations. In the camera-ready version we will replace it with:
>
>     "Table 10 shows that (a) recent architectures already reduce HD error relative to older baselines, and (b) adding MG yields a further 15-25% reduction, highlighting MG's specific contribution to non-stationarity mitigation."
>
> This revision explicitly separates the effects of modern backbones from the additional gains brought by MG, eliminating the contradiction.
>
> ### **Additional Clarification about Q7 & Q8 & L5**
>
> We confirm that every wording adjustment highlighted by the reviewer will be incorporated into the camera-ready version. Moreover, all baselines in our experiments were executed with their original default hyper-parameters as released in the corresponding papers or official repositories. No extra tuning was applied, ensuring that the observed performance gains arise exclusively from the proposed Meta Guidance mechanism.

---

> ### Author Response · Authors · 2025-08-04
> **Look forward to your feedback**
>
> Dear Reviewer NQxZ,
>
> Thank you very much for your thorough and insightful comments, and for recommending our paper for borderline accept. In our previous rebuttal we have already addressed every point you raised:
>
> - **Broader Scope to Missing Patterns** (W1, L2, L4) – new time-varying, and consecutive-gap experiments.
>
> - **Dataset Suitability for Hyper-parameter Study** (W2, W8, Q6) – additional $k$ and $r$ sweeps on Weather, Traffic, and HD showing robustness.
>
> - **Impact of $\lambda$** (W3, Q5) – new "MG w/o $\lambda$" ablation highlighting its necessity.
>
> - **Code Availability** (W4, L5) – anonymous repository link now public.
>
> - **Baseline Context** (W5) – concise table to be added in the revision.
>
> - **Synthetic-Data Evaluation** (W6, L1) – preliminary synthetic-signal study confirming MG's gains.
>
> - **Simple Baseline Comparisons** (W7, L3) – Linear, KNN, and Seasonal baselines now included.
>
> - **Clarifications** (Q1-Q4) – clarified MG/PG's novelty, quantified consistent gains on strong backbones, justified rr, and fixed ambiguous wording.
>
> We would be happy to provide further responses. Look forward to your feedback.
> Thanks for your comment again!
>
> Best regards,
>
> All Authors

---

> ### Comment · Reviewer_NQxZ · 2025-08-07
>
> Thank you for your detailed explanations and the additional experiments/results. I appreciate the time you've taken to address these points, and I apologize for my late response.
>
> ## Hyper-parameter Study Observations
>
> Regarding the top-k frequency selection method, I'm somewhat surprised by the results. Based on my understanding, I would have expected high seasonal datasets to show improvement as $k$ increases. However, this doesn't appear to be the case in your results. This makes me wonder whether:
> - Top-k might not be the optimal method for selecting important frequencies, or
> - I might be missing something in my interpretation of the results
>
> Could you please provide additional clarification on this aspect?
>
> ## Implementation of Simple Baselines
>
> I was somewhat surprised that the Seasonal Historical Average baseline didn't perform better. To better understand these results, could you please explain how you implemented this baseline? Unfortunately, I'm still unable to access the Git repository (the page remains stuck on the "loading..." screen and the I have an "error" screen), which prevents me from examining the implementation details directly (if you provided it).
>
> ## Presentation of Marginal Gains
>
> If you wish to emphasize the gains achieved by your proposal, I would recommend computing the metrics using non-normalized values. This approach differs from what is typically done in most papers, but it could potentially make the improvements more apparent and impactful to readers.
>
>
> At this point, I don't have any additional questions as I'm satisfied with the answers provided. However, I was interested in seeing reviewer i1nN's comments on your rebuttal.
>
> Additionally, regarding reviewer i1nN's comment:
> > There is no comparison against other approaches handling [...] periodicity.
>
> I believe they were suggesting comparisons of your Periodic Guidance (PG) component against other models that specifically address periodicity, such as:
> - CycleNet [1]
> - PerimidFormer [2]
> - FRNet [3]
>
> These comparisons could provide valuable context for evaluating the effectiveness of your periodicity handling approach.
>
> [1] Lin, et al. "Cyclenet: Enhancing time series forecasting through modeling periodic patterns."
>
> [2] Wu, et al. "Peri-midformer: Periodic pyramid transformer for time series analysis."
>
> [3] Zhang, et al. "FRNet: Frequency-based Rotation Network for Long-term Time Series Forecasting."

---

> > ### Author Response · Authors · 2025-08-08
> >
> > We sincerely thank the reviewer for taking the time to re-engage with our rebuttal and for providing further constructive feedback. We are pleased to address the points raised below.
> >
> > # 1. Clarification on the Top-k Frequency Selection Results
> >
> > We understand the reviewer’s expectation that high-seasonality datasets might benefit more as $k$ increases. we carried out a frequency-domain energy analysis on both the Weather and Traffic datasets (as recommended in Q6). The results are presented below.
> >
> > ||top 1|top 2|top 3|top4
> > -|-|-|-|-
> > Weather|69.87%|75.42%|77.22%|78.18%
> > Traffic|75.68%|79.89%|82.24%|83.19%
> >
> > These results reveal a key insight:
> > **The dominant periodic patterns are captured by only a few strong spectral peaks.**
> >
> > **Dominance of the top few peaks**: On datasets like Weather and Traffic, the dominant seasonal cycles (e.g., daily and weekly patterns) are represented by a small number of sharp peaks in the frequency spectrum. Beyond these, additional frequencies contribute marginal energy (e.g., only ~1% gain from top-3 to top-4 in both datasets). This shows that most useful periodic information is already captured by the first few peaks, and adding more tends to introduce harmonics or low-energy noise components with little practical benefit.
> >
> > # 2. Implementation Details of the Seasonal Historical Average Baseline
> >
> > Our Seasonal Historical Average baseline follows a **widely used seasonal-slot averaging strategy** in time-series imputation. The idea is to group data points by recurring seasonal keys (in our case, **month** and **hour**) and use the mean of all observed values in the same slot to fill a missing value.
> >
> > For example, in the Traffic dataset, if the data point for **Aug 1, 10:00** is missing, we look at all other observed values recorded at 10:00 in August—such as **Aug 2, 10:00** and **Aug 3, 10:00**—and take their average as the imputed value for Aug 1, 10:00. This process is repeated independently for each channel.
> >
> > This approach is simple, interpretable, and dataset-agnostic. However, **due to time constraints**, our current implementation uses only the fixed (month,hour) seasonal key and does not adapt to dataset-specific periodicities (e.g., weekly cycles in traffic or weather datasets). As a result, its performance can be limited when:
> >
> > - There is **high variability within the same seasonal slot** (e.g., unusual events on a given day).
> >
> > - The **true dominant cycle** does not align with the fixed month–hour granularity.
> >
> > We acknowledge this as a limitation of our current implementation. In future work, we plan to enhance this baseline by **automatically detecting dominant cycles** and incorporating **multi-scale seasonal keys** to better align with the underlying periodic structures of each dataset.
> >
> > # 3. On Presenting Gains with Non-Normalized Metrics
> >
> > We appreciate the suggestion. While we follow the standard practice of normalizing inputs and reporting results on the normalized scale for fair comparison across backbones, we agree that presenting metrics in absolute scale can make the improvements more tangible to readers. However, evaluating training and inference under truly unnormalized conditions would require re-running all main experiments from scratch which is computationally expensive.
> >
> > Due to **time constraints during the rebuttal phase**, we were unable to retrain all models on unnormalized scales. As an initial step, we computed denormalized MAE for several representative datasets, as shown below:
> >
> > ||HD|Weather|ETTh1
> > -|-|-|-
> > Transformer|876696|9.948|1.139
> > Transformer+MG|169067|4.252|0.561
> >
> > These results clearly demonstrate the substantial gains achieved by MG in original units. In the camera-ready version, we will include a comprehensive table of non-normalized results for all datasets in the appendix to provide a complete overview.
> >
> > # 4. Comparisons Against Other Periodicity-Specific Models
> >
> > We thank the reviewer for suggesting periodicity-focused works such as CycleNet, Peri-midformer, and FRNet.
> >
> > - **CycleNet and FRNet are designed specifically for long-term time-series forecasting**, and neither the papers nor their released code address imputation tasks. As such, their objectives are not directly aligned with our work, which focuses on multivariate time-series imputation.
> >
> > Adapting these models to imputation requires non-trivial architectural modifications beyond simple retraining. Due to **time constraints during the rebuttal phase**, we were unable to fully reimplement and evaluate CycleNet and FRNet in the imputation setting. However, Peri-midformer can be more readily adapted to imputation. Following the reviewer's suggestion, we conducted additional experiments by integrating Peri-midformer into our evaluation:
> >
> > ||ETTh1|ETTm1
> > -|-|-|
> > Peri-midformer|0.174|0.108
> > CSDI+MG|0.100|0.065
> >
> > In the camera-ready version, we will adapt CycleNet and FRNet to the imputation task and include all three models as baselines in an expanded comparison to further contextualize PG.

---

> ### Comment · Reviewer_NQxZ · 2025-08-08
>
> Thank you for providing these additional explanations and results.
>
> ## Regarding Periodicity-Specific Models
>
> While I'm not highly familiar with FRNet, I understand that CycleNet is designed to learn the main cycles in a dataset and has a similar usage to iTransformer. Given this similarity, it seems that using CycleNet shouldn't be significantly more difficult than how you've already incorporated iTransformer.
>
> ## Questions About Top-k Frequency Analysis
>
> Let me summarize my understanding to ensure I've grasped the concept correctly:
> - The frequency domain analysis identifies the importance of each top frequency for the dataset in question
> - The previous table showing results with varying k (from 1 to 20) demonstrates your proposal's performance across different numbers of top frequencies included
>
> If this understanding is correct, I notice that while there's approximately a 10% difference in importance between the top-1 and top-4 frequencies, your proposal's performance actually decreases as we consider more frequencies. This seems counterintuitive given that:
> > most useful periodic information is already captured by the first few peaks
>
> However, your proposal doesn't appear to benefit from the important information provided by the second, third, and fourth peaks, even though they make significant contributions. Could you provide any interpretation or explanation for this observation?
>
> ## Regarding Non-Normalized Performance
>
> I can't access your repository, so it is difficult to judge; but usually:
> 1. Input data are normalized before passing it to the model
> 2. Normalized outputs are then obtained from the model
> 3. (Most papers) Calculate metrics on these normalized values (which, in my opinion, don't fully reflect real-world performance)
> But it is not that complicated to inverse normalized the obtained values, compare them to the actual and report metrics on these unnormalized comparisons.
>
> I'm somewhat confused by the authors' mention of needing to retrain models for this purpose. Models like iTransformer typically include an `inverse_transform` function in their `data_loader`, it should be possible to obtain these unnormalized performance metrics without significant computational expense.
>
> In any case, it is just a suggestion to make the performance of your proposal more visually impactful and meaningful for readers, I understand if you choose not to address this in the current phase of revision.

---

> > ### Author Response · Authors · 2025-08-09
> >
> > We sincerely thank the reviewer for taking the time to re-engage with our rebuttal and for providing further constructive feedback. We are pleased to address the points raised below.
> >
> > ## 1. Regarding Periodicity-Specific Models (CycleNet)
> >
> > We thank the reviewer for noting CycleNet's similarity to iTransformer. Following this suggestion, we conducted a preliminary adaptation of CycleNet for imputation:
> > |    | ETTh1     | ETTm1     |
> > | ------------------ | --------- | --------- |
> > | CycleNet       | 0.219     | 0.159     |
> > | Peri-midformer | 0.174     | 0.108     |
> > | CSDI+MG        | 0.100 | 0.065 |
> >
> > However, this quick adaptation may not fully exploit CycleNet's potential for imputation due to task-specific tuning needs. We will perform a more thorough adaptation and evaluation of CycleNet and FRNet in next version to ensure fair and optimized comparisons.
> >
> > ## 2. Clarification on the Top-k Frequency Selection Results
> >
> > We appreciate the reviewer's careful observation. Based on our analysis, we believe two main factors may help explain why including more top frequencies does not consistently yield better results:
> >
> > - **Frequency distortion from pre-filling**: As ours is an imputation task, we first apply linear interpolation before FFT to obtain a complete sequence. While simple and efficient, this step can **alter the original spectral distribution**.  In the Traffic dataset, for example, FFT generally captures the first few dominant periods accurately, but when the number of significant periods increases, distortions become more likely. For instance, the frequency of the fourth-largest peak shifts substantially—from 0.0049 in the original data to 0.125 after introducing 25% missingness and applying linear interpolation. Such shifts suggest that i**ncreasing $k$ may introduce distorted or spurious periodic components rather than true underlying cycles**, which can lead to unstable performance.
> >
> > - **Backbone's own spectral learning**: The backbone models (e.g., Transformer, TimesNet) can already capture weaker seasonal components internally. Thus, once PG provides the most salient cycles, adding more has diminishing or neutral effects.
> >
> >
> > ## 3. On Non-Normalized Performance Reporting
> >
> > We appreciate the reviewer's insightful point. We fully agree that normalized and non-normalized evaluations serve different purposes:
> >
> > - Normalization is typically applied to unify the scale across channels and ensure that each channel contributes equally to the loss.
> >
> > - Non-normalized evaluation reflects raw-unit performance, but in such settings, channels with larger numerical ranges can dominate aggregate metrics such as MAE.
> >
> >
> > Following the reviewer's suggestion, we applied the inverse_transform approach to the outputs of all datasets without retraining the models. The results (MAE) are shown below:
> >
> > | Model              | HD          | Weather   | ECL         | Traffic   | ETTh1     | ETTh2     | ETTm1     | ETTm2     | TCPC        |
> > | ------------------ | ----------- | --------- | ----------- | --------- | --------- | --------- | --------- | --------- | ----------- |
> > | Transformer        | 522625     | 5.609     | 244.2     | 0.010     | 0.814     | 1.518     | 0.609     | 1.266     | 465.9     |
> > | Transformer+MG | 169737 | 4.015 | 175.1 | 0.009 | 0.572 | 0.708 | 0.350 | 0.465 | 169.8 |
> >
> >
> > When compared with the non-normalized results from our previous reply (e.g., Transformer on HD: **522625** here vs. **876696** in the direct raw-data experiment), we can better explain our earlier choice to retrain. Intuitively, to obtain performance metrics on the original data scale, the model should be trained with raw inputs—**normalizing the data changes the training dynamics, as channels with larger magnitudes would otherwise dominate the loss**. Our new experiments confirm that this difference in preprocessing leads to different outcomes. Interestingly, they also show that applying normalization before training can in fact improve performance, which suggests that the reviewer's proposed inverse_transform approach is, in some cases, a more favorable option. We will therefore include the complete inverse_transform results table in the camera-ready version to better reflect real-world performance.
> >
> > We sincerely appreciate the reviewer's thoughtful suggestion, which not only strengthens our analysis but also improves the clarity and practical relevance of our reported results.
> >
> > Regarding the repository access issue, we have confirmed with the relevant authors that the provided link is accessible on their side. If you could kindly share more details (e.g., error message, network environment), we would be happy to investigate further and ensure you can access all materials without difficulty.

---

### Note · Authors · 2025-08-12

Dear AC,

We sincerely thank the AC and the reviewers for their constructive feedback and fruitful discussions.

# Key clarifications & resolutions:
- **Scope & robustness**. Beyond MCAR/MAR/MNAR, we evaluate time-varying missingness and consecutive gaps; MG remains robust across these settings.

- **Baselines & fairness**. We add simple preprocessing (detrending) and classical imputers (linear/KNN/seasonal), strong forecasters/backbones (PatchTST, Non-Stationary Transformer), and representative model-based imputers (TimeCIB, GP-VAE, FGTI). Across these, MG achieves the strongest overall results.

- **Ablations & stability.** Removing the meta-weight $\lambda$ (simple averaging) degrades performance, supporting adaptive weighting. Sweeps over k and r show stable behavior and low tuning burden; making r, sigma learnable provides little benefit.

- **Design choice.** Channel-independent guidance outperforms a channel-averaged variant and does not harm multivariate imputation.

- **Generalization & utility.** On additional datasets (PhysioNet, PM2.5), gains persist. Using MG-imputed series improves downstream forecasting (e.g., TimesNet/iTransformer on Weather).

- **Efficiency & clarity.** MG retains linear-time complexity and incurs only small wall-clock overhead. We clarify tabular/figure semantics and provide an accessible anonymous repository.

# Strengths unanimously recognized by reviewers:
- Reviewers highlighted the paper's **novelty** (i1nN), **clear structure and readability** (i1nN, bot2), and that MG is **plug-and-play** (NQxZ, sQEU, bot2), effective across **diverse backbones and datasets** (NQxZ, sQEU, bot2), and consistently improves performance under realistic missingness (sQEU, bot2)—indicating strong generalization ability and practical value.

# Assessment requested
- One post-rebuttal comment (i1nN) concerns whether simple detrending matches MG and whether MG's overhead is justified.

Our analyses show that MG consistently outperforms detrending (paired tests, p = 0.011), achieving >10% relative MAE reductions vs. detrending. It preserves $\mathcal{O}(T)$ complexity and has runtime comparable to detrending; empirically, MG's overhead is <1.5% of the backbone's wall-clock time. As the reviewer did not follow up, we respectfully request that the AC **reassess this concern in light of the submitted evidence**.

Best regards,

Authors

---

### Decision · Program_Chairs · 2025-09-17

**Decision:**

Accept (poster)

**Comment:**

This work introduces a plug-and-play module named Meta Guidance (MG), which improves multivariate time series imputation by combining Non-Stationary Guidance (NSG) and Periodic Guidance (PG) mechanisms. The method uses a learnable parameter λ to dynamically weight guidance from nearby temporal measurements versus corresponding periodic timesteps. MG integrates seamlessly with existing architectures and achieves consistent 27% performance improvements across nine datasets.

The plug-and-play design enables broad compatibility with existing imputation methods (from Reviewer NQxZ, Reviewer sQEU, Reviewer bot2). Extensive experiments demonstrate consistent improvements across multiple datasets, missing mechanisms, and baseline comparisons (from Reviewer NQxZ, Reviewer i1nN, Reviewer sQEU, Reviewer bot2).

There are some initial comments from reviewers: (1) Missing comparisons with other non-stationarity/periodicity approaches and key baselines like PatchTST and Non-stationary Transformers (from Reviewer i1nN, Reviewer bot2). (2) The channel-independent design may mislead imputation for datasets requiring inter-channel dependencies (from Reviewer i1nN). (3) Limited analysis of the critical λ parameter's impact across different datasets (from Reviewer NQxZ, Reviewer i1nN). The method focuses on individual missing points rather than consecutive missing patterns common in real scenarios (Reviewer NQxZ).

Authors have addressed most of them, and reviewers acknowledge that the author responses are informative.